# Dynamic Neural Graph Encoding of Inference Processes in Deep Weight Space

**Di Wu**  *E1538518@u.nus.edu*
*University of Toronto*
*National University of Singapore*

**Huan Liu**[*]  *liuh127@outlook.com*
*McMaster University*

**Zhixiang Chi**
*University of Toronto*

**Yuanhao Yu**
*McMaster University*

**Konstantinos N. Plataniotis**
*University of Toronto*

**Yang Wang**
*Concordia University*

**Reviewed on OpenReview:** *https://openreview.net/forum?id=4fweEyVYLF*

## Abstract

The rapid advancements in using neural networks as implicit data representations have attracted significant interest in developing machine learning methods that analyze and process the weight spaces of other neural networks. However, efficiently handling these high-dimensional weight spaces remains challenging. Existing methods often overlook the sequential nature of layer-by-layer processing in neural network inference. In this work, we propose a novel approach using dynamic graphs to represent neural network parameters, capturing the temporal dynamics of inference. Our Dynamic Neural Graph Encoder (DNG-Encoder) processes these graphs, preserving the sequential nature of neural processing. Additionally, we also leverage DNG-Encoder to develop INR2JLS (Implicit Neural Representation to Joint Latent Space) for facilitate downstream applications, such as classifying Implicit Neural Representations (INRs). Our approach demonstrates significant improvements across multiple tasks, surpassing the state-of-the-art INR classification accuracy by approximately 10% on the CIFAR-100-INR. Our code is available at `https://github.com/dddiowww/DNG`.

## 1 Introduction

Deep neural networks have demonstrated superb capability in addressing real-world problems in fields such as computer vision, natural language processing, and the natural sciences. While it is generally used for learning patterns from data, recent studies have expanded its scope by treating neural networks themselves as inputs, enabling tasks such as optimizing networks (Metz et al., 2022), predicting the labels of the data encoded in implicit neural representations (Dupont et al., 2022), and generating or modifying their weights to alter functionality (Schürholt et al., 2022). However, processing these weight spaces presents considerable challenges due to their complex, high-dimensional nature.

---

[*]Research Lead. Corresponding Author.

To address the difficulty, some existing methods propose to narrow the effective weight space using a restricted training process (Bauer et al., 2023; De Luigi et al., 2023). However, this neglects the crucial permutation symmetry property of the neural network weights, *i.e.,* neurons within a layer can be rearranged without altering the network's function (Hecht-Nielsen, 1989). Overlooking the permutation symmetry can significantly increase the search space for optimal parameters of processing network, resulting in reduced generalization and unsatisfactory performance. By observing this, recent works (Navon et al., 2023; Zhou et al., 2024a;b) build permutation equivariant weight-space models named *neural functionals*. Unfortunately, these methods need manual adaptation for each new architecture, and a single model can only handle one fixed architecture. To enable processing heterogeneous architectures, Kofinas et al. (2024) and Lim et al. (2024) introduce to model neural network weights as graphs, which links neural network parameters similarly to a computation graph. These methods, while innovative, predominantly employ static graphs. This static representation allows GNNs to process the entire graph in a single pass. However, such an approach overlooks a critical aspect of neural network behavior during inference: the sequential nature of layer-by-layer processing. Neural networks, by design, perform inferences in a temporally ordered manner. This sequential dependency suggests that a more natural and effective modeling of neural network parameters could be achieved by dynamic graphs. Unlike static graphs, dynamic graphs evolve over time, capturing the temporal dynamics inherent in the forward pass process.

Motivated by these observations, we propose a novel method that represents neural network parameters as dynamic graphs, namely dynamic neural graphs. Leveraging this dynamic graph, we introduce the Dynamic Neural Graph Encoder (DNG-Encoder), a recurrent-like graph neural network designed to process dynamic neural graphs. This approach mirrors the forward propagation mechanism of neural networks, preserving the temporal characteristics of the data flow through the layers. To facilitate downstream applications, we use the DNG-Encoder to develop INR2JLS, a method that learns a joint latent space between deep weights and the original data. This approach provides a more informative latent space compared to previous methods that focused solely on deep weight space.

Our contribution can be summarized as follows. First, we introduce the concept of dynamic neural graphs for modeling neural network parameters, capturing the temporal dynamics of the forward pass. Second, we develop a novel RNN-based graph neural network to process these dynamic graphs, effectively imitating the sequential nature of neural network inference. Third, we propose INR2JLS, a technique that maps INR weights into a joint latent space that can benefit downstream applications. Finally, we show through extensive experiments the effectiveness of our method across three tasks. Notably, the performance of our method improves over the state-of-the-art by 9% and 10% on CIFAR-10 and CIFAR-100 for INR classification.

## 2   Related Work

**Dynamic Graphs.** Dynamic graphs are categorized into discrete-time (DTDGs) and continuous-time (CTDGs) types. DTDGs consist of graph snapshots at regular intervals, each treated as a static graph, while CTDGs evolve through an initial static state and a sequence of timestamped events. To capture both structural and temporal information, dynamic graph neural networks have been widely studied, often employing an encoder-decoder framework (Kazemi et al., 2020), where the encoder learns node embeddings and the decoder performs downstream tasks. A key approach to handling dynamic graphs involves using recurrent neural networks to model temporal dependencies. Notable methods for DTDGs include Seo et al. (2018); Chen et al. (2022), while CTDGs have been addressed by Trivedi et al. (2017; 2019).

**Implicit Neural Representations.** Recent works employ a neural network as a continuous function to implicitly represent the objects or shapes (Genova et al., 2019; Gropp et al., 2020). This function takes an input (often a point coordinate) and outputs the corresponding feature value or property. SIREN (Sitzmann et al., 2020) is a continuous implicit neural representation that utilizes sine as a periodic activation function. It excels at fitting complex signals, including natural images and 3D shapes. In our experiments, the INRs in the datasets we used are in the form of SIRENs.

**Learning in Deep Weight Spaces.** Weight Space Learning has recently been formalized as a unified paradigm that treats neural network parameters as a structured data modality (Han et al., 2026b). The high-dimensional weight spaces of implicit neural representations (INRs) pose challenges for extracting meaningful

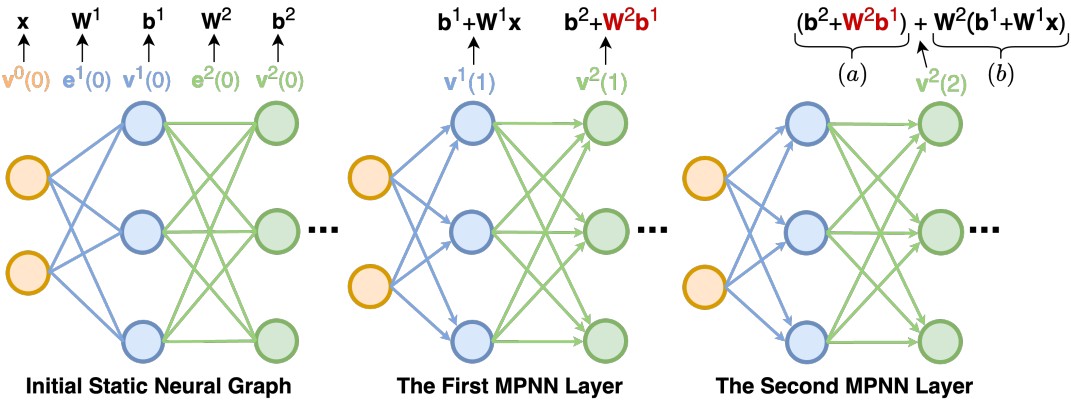

Figure 1: An illustration of the limitations of static neural graph processing. In deeper layers, updated nodes may incorporate undesired information, such as $\mathbf{W}^2\mathbf{b}^1$ in (a).

information. Some works address this by constraining training to narrow the effective weight space (Bauer et al., 2023; De Luigi et al., 2023), but Navon et al. (2023) and Zhou et al. (2024a) highlight that such methods often overlook permutation symmetry in neural weights, which can hinder generalization. To mitigate this, they proposed permutation-equivariant weight-space models, with further performance enhancement via transformer-based architectures (Zhou et al., 2024b). Beyond permutation, recent research such as W2T (Han et al., 2026a) underscores the necessity of resolving inherent reparameterization symmetries through canonical representations to accurately decode model behavior from weights. However, these methods require manual adaptation, limiting scalability. To generalize across architectures, Kofinas et al. (2024) and Lim et al. (2024) model neural weights as graphs, leveraging graph neural networks (GNNs) for a symmetry-aware and expressive approach. While effective, these methods rely on static graphs and positional encoding to capture layer hierarchy (Andreis et al., 2024). In this paper, we extend this paradigm by incorporating the temporal nature of neural inference. We propose transforming neural networks into dynamic graphs and applying temporal graph neural networks to better capture inter-layer dependencies.

## 3 Preliminaries on Neural Graph

### 3.1 Neural Networks as Static Neural Graph

Neural graphs (Kofinas et al., 2024) have been introduced as a graph-like representation of neural network weights. For example, in an L-layer multilayer perceptron (MLP) $\mathbf{M}$, the weight matrices are denoted as $\{\mathbf{W}^1, \mathbf{W}^2, ..., \mathbf{W}^L\}$, and the biases are denoted as $\{\mathbf{b}^1, \mathbf{b}^2, ..., \mathbf{b}^L\}$. Each weight matrix $\mathbf{W}^l$ and bias $\mathbf{b}^l$ respectively have dimensions $d^l \times d^{l-1}$ and $d^l$. $\mathbf{M}$ can be converted to a neural graph $\mathcal{G} = (\mathbf{V}, \mathbf{E})$, where $\mathbf{V} = \{\mathbf{v}^0, \mathbf{v}^1, ..., \mathbf{v}^L\}$ denotes the set of nodes features, and $\mathbf{E} = \{\mathbf{e}^1, \mathbf{e}^2, ..., \mathbf{e}^L\}$ represents the set of edges features. $\mathbf{v}^l \in \mathbb{R}^{d^l \times d_v}$ and $\mathbf{e}^l \in \mathbb{R}^{d^l \times d^{l-1} \times d_e}$ are the nodes and edges of the $l$-th layer in $\mathcal{G}$, where $d_v$ and $d_e$ represent the dimensions of the node feature and the edge feature. In addition, $\mathbf{v}^l$ and $\mathbf{e}^l$ correspond to neurons at the $l$-th layer of $\mathbf{M}$ and connections between neurons at the $l$-th layer and the $(l-1)$-th layer of $\mathbf{M}$. Typically, edge features contain matrix weights, while node features are constructed using biases. Frequency representations, such as Random Fourier Features (RFFs), have shown improved performance on weights and biases (Zhou et al., 2024a;b). Notably, once defined, the graph structure remains fixed, thus constituting a *static* neural graph.

### 3.2 Expressivity of Processing Neural Graphs with Graph Neural Networks (GNNs)

A recent study (Navon et al., 2023) suggests that if a model can simulate the forward pass of its input neural network, it can match that network's expressive power. Following this, Kofinas et al. (2024) employs GNNs in the form of message passing neural networks (MPNNs) (Gilmer et al., 2017).

Consider an $L$-layer network $\mathbf{M}$ with input $\mathbf{x} \in \mathbb{R}^{d^0}$. Its first-layer activation for neuron $i$ is

$$\mathbf{a}_i^1 = \sigma(\mathbf{b}_i^1 + \sum_j \mathbf{W}_{ij}^1 \mathbf{x}_j). \tag{1}$$

Meanwhile, an MPNN with $K$ layers (often $K = L$) processes node and edge features $\mathbf{v}^l(k)$ and $\mathbf{e}^l(k)$. The first MPNN layer's message-passing update is

$$\mathbf{v}_i^1(1) = \phi_u^1 \left( \mathbf{v}_i^1(0), \sum_{j \in N_i} \phi_m^1(\mathbf{v}_i^1(0), \mathbf{e}_{ij}^1(0), \mathbf{v}_j^0(0)) \right), \tag{2}$$

where $i$ and $j$ represent the index of the target node and the source node. $N_i$ represents the neighbors of node $\mathbf{v}_i$. Please note that $\mathbf{v}_i^1(0)$, $\mathbf{e}_{ij}^1(0)$, and $\mathbf{v}_j^0(0)$ are the initial representations directly derived from $\mathbf{b}_i^1$, $\mathbf{W}_{ij}^1$, and $\mathbf{x}_j$, respectively. For simplicity, here we assume these indices match thoes of network parameters.

By comparing the Equation 1 and 2, it can be found that the MPNN can approximate the feed-forward procedures on input networks of the first layer in MPNN as follows. The function $\phi_m^1$ is the message function of the first MPNN layer, capable of approximating the product $\mathbf{W}_{ij}^1 \mathbf{x}_j$. The function $\phi_u^1$ represents the node update function of the first MPNN layer. It can easily approximate the operation of adding $\mathbf{b}_i^1$ to $\mathbf{W}_{ij}^1 \mathbf{x}_j$ and applying the activation function $\sigma$. Thus, the MPNN can effectively approximate the feed-forward process of the input network.

### 3.3 Limitations of Static Neural Graphs

In standard neural networks, inference proceeds sequentially: each layer depends on the outputs of all prior layers. By contrast, in a static neural graph, each node can only aggregate information from adjacent layers, contradicting the true layer-by-layer dependency. Moreover, we also identify a significant issue when employing multi-layer graph neural networks discussed below.

Following the message-passing process of the first MPNN layer on the first set of MLP nodes in Equation 2, the message-passing process of the first MPNN layer on the second set of MLP nodes is as $\mathbf{v}_i^2(1) = \phi_u^1(\mathbf{v}_i^2(0), \sum_{j \in N_i} \phi_m^1 (\mathbf{v}_i^2(0), \mathbf{e}_{ij}^2(0), \mathbf{v}_j^1(0)))$. Since this graph update uses the same $\phi_u^1$ and $\phi_m^1$ as those in Equation 2, according to the expressivity of GNN, each node $\mathbf{v}_i^2(1)$ in the second layer of the MLP should be updated to $\mathbf{b}_i^2 + \mathbf{W}_i^2 \mathbf{b}^1$. For clarity, we omit including $\sigma$.

Now, we move to the second layer of MPNN. When we examine the update of node $\mathbf{v}_i^2(2)$ in this layer, we have: $\mathbf{v}_i^2(2) = \phi_u^2(\mathbf{v}_i^2(1), \sum_{j \in N_i} \phi_m^2(\mathbf{v}_i^2(1), \mathbf{e}_{ij}^2(1), \mathbf{v}_j^1(1)))$. Again, we can assume this calculation completes: $\underbrace{\mathbf{b}_i^2 + \mathbf{W}_i^2 \mathbf{b}^1}_{(a)} + \underbrace{\mathbf{W}_i^2(\mathbf{b}_i^1 + \mathbf{W}_i^1 \mathbf{x})}_{(b)}$, where part (b) is approximated by $\phi_m^2$, and adding part (a) to (b) is done by $\phi_u^2$. However, if we strictly follow the forward pass of neural network, the desired computation should be: $\underbrace{\mathbf{b}_i^2}_{(c)} + \underbrace{\mathbf{W}_i^2(\mathbf{b}_i^1 + \mathbf{W}_i^1 \mathbf{x})}_{(d)}$. In this case, besides the approximation of addition operation, $\phi_u^2$ needs to do additional work to extract $\mathbf{b}_i^2$ from the summarized result $\mathbf{b}_i^2 + \mathbf{W}_i^2 \mathbf{b}^1$. This may seem like an easy task, but technically it is not. Isolating $\mathbf{b}_i^2$ from $\mathbf{b}_i^2 + \mathbf{W}_i^2 \mathbf{b}^1$ is an ill-posed inverse problem, making training difficult and often leading to suboptimal solutions. We show the problem in Figure 1 and further illustrate it with a scalar example below.

**Scalar example.** Consider a 2-layer MLP with scalar parameters $w^1=3$, $b^1=2$, $w^2=5$, $b^2=1$, and input $x$. The true forward pass (omitting $\sigma$ for clarity) computes $y = b^2 + w^2(b^1 + w^1 x) = 1 + 5(2 + 3x)$. After the first MPNN layer on the corresponding static neural graph:

$$v^1(1) = b^1 + w^1 x = 2 + 3x \qquad \text{(correct first-layer activation)},$$
$$v^2(1) = b^2 + w^2 b^1 = 1 + 10 = 11 \qquad \text{(undesired term } w^2 b^1 = 10).$$

In the second MPNN layer, $\phi_u^2$ receives $v^2(1) = 11$ and the message $\text{msg} \approx w^2(b^1 + w^1 x) = 5(2+3x)$. A natural update, adding the node state to the message, yields $11 + 5(2+3x)$, which exceeds the desired result

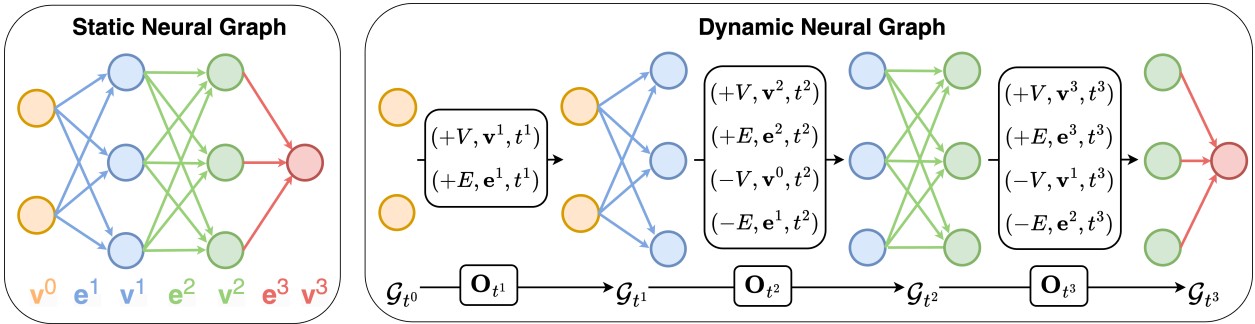

Figure 2: Left: static neural graph. Right: dynamic neural graph. A static neural graph has a fixed structure and set of connections, while dynamic neural graph evolves over time, with changes in its structure and node connections.

$1 + 5(2+3x)$ by exactly $w^2 b^1 = 10$. Correcting this requires $\phi_u^2$ to recover $b^2 = 1$ from the compound value $v^2(1) = 11 = b^2 + w^2 b^1$, but $w^2 b^1$ is not separately available as an input. Across different input MLPs, $w^2 b^1$ takes arbitrary values, so no fixed function of the scalar 11 can isolate $b^2$ in general.

## 4 Neural Networks as Dynamic Graph

To address the limitations of static neural graphs, we convert neural networks into dynamic graphs. By integrating the inherent temporal processing characteristics of neural networks into their graph representation, subsequent models can more effectively capture and utilize these temporal relationships. In the following sections, we discuss the conversion of both MLPs and CNNs into dynamic neural graphs.

### 4.1 MLPs as Dynamic Neural Graphs

We define a dynamic graph converted from an $L$-layer MLP $\mathbf{M}$ as a *dynamic neural graph* $\mathcal{G}_T = (\mathcal{G}_{t^0}, \mathbf{O}_{[t^1:t^L]})$, where $\mathcal{G}_{t^0}$ only contains $\mathbf{v}^0$ that corresponds to inputs of $\mathbf{M}$. The definitions of $\mathbf{v}^l$ and $\mathbf{e}^l$ in the dynamic neural graph are the same as those in the static neural graph from Section 3.1. Since the inputs to neural networks are not fixed, we treat them as learnable vectors. To keep the dimensions of all node embeddings consistent, we set their dimensions to be the same as the dimensions of the embeddings of other nodes. We simulate the forward pass process of $\mathbf{M}$ by defining the graph update event $\mathbf{O}$. We define four graph operations, *i.e.,* edge addition $(+E)$, edge deletion $(-E)$, node addition $(+V)$ and node deletion $(-V)$. Specifically, a graph update event $\mathbf{O}_{t^l}$ at time $t^l$ is:

$$\mathbf{O}_{t^l} = \begin{cases} \{(+V, \mathbf{v}^l, t^l), (+E, \mathbf{e}^l, t^l)\}, & \text{if } l = 1, \\ \{(+V, \mathbf{v}^l, t^l), (+E, \mathbf{e}^l, t^l), \\ \quad (-V, \mathbf{v}^{l-2}, t^l), (-E, \mathbf{e}^{l-1}, t^l)\}, & \text{if } 1 < l \leq L, \end{cases} \tag{3}$$

where $(+V, \mathbf{v}^l, t^l)$ denotes adding the nodes $\mathbf{v}^l$ to $\mathcal{G}_{t^l}$ at $t^l$. $(+E, \mathbf{e}^l, t^l)$ represents adding edges $\mathbf{e}^l$ to to $\mathcal{G}_{t^l}$ at timestamp $t^l$. The edges $\mathbf{e}^l$ connect nodes $\mathbf{v}^{l-1}$ to the newly added nodes $\mathbf{v}^l$. When $t^1 < t^l \leq t^L$, we delete nodes $\mathbf{v}^{l-2}$ and edges $\mathbf{e}^{l-2}$, and adding incoming nodes and edges.

Similar to many previous approaches to process weight space parameters (Zhou et al., 2024a;b; Kofinas et al., 2024), we use the Random Fourier Features (RFFs) of weights $\mathbf{W}^l$ and biases $\mathbf{b}^l$ in $\mathbf{M}$ to initialize $\mathbf{v}^l$ and $\mathbf{e}^l$. By the above definition of $\mathbf{O}$, the snapshot $\mathcal{G}_{t^l} = (\{\mathbf{v}^{l-1}, \mathbf{v}^l\}, \mathbf{e}^l)$ at timestamp $t^l$ can be considered a fully connected static bipartite directed graph (Bang-Jensen & Gutin, 2008). This graph consists of two sets of nodes, $\mathbf{v}^{l-1}$ and $\mathbf{v}^l$, with the direction of edges $\mathbf{e}^l$ from nodes $\mathbf{v}^{l-1}$ to nodes $\mathbf{v}^l$. In this way, the structure of $\mathcal{G}_{t^l}$ complies with the topology of the $l$-th forward pass step of $\mathbf{M}$. To illustrate the procedure of converting an MLP to a dynamic neural graph, we show an example in Figure 2.

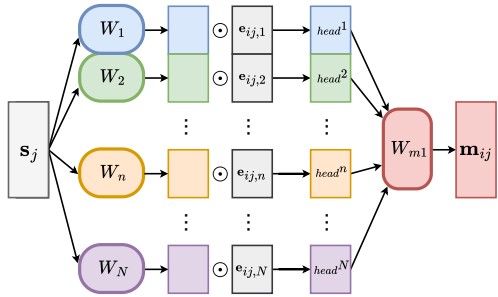

Figure 3: An illustration of multi-head message function, formalized in Equation 5. Each edge receives a distinct message from the source node via separate heads. The aggregated message is computed by merging all heads with an MLP.

## 4.2 CNNs as Dynamic Neural Graphs

Here, we introduce the procedure to convert convolutional layers and linear layers in CNNs to modules in dynamic neural graphs. [1]

A 2D convolutional layer at the $l$-th level of the CNN includes filter $\mathbf{W}^l \in \mathbb{R}^{c^l \times c^{l-1} \times h^l \times w^l}$ and bias $\mathbf{b}^l \in \mathbb{R}^{c^l}$, where $c^{l-1}$ and $c^l$ respectively denote the depth and the number of filters in the $l$-th convolutional layer. Typically, $c^{l-1}$ strictly matches the number of input channels, and $c^l$ controls the number of output channels. $h^l$ and $w^l$ represent the width and height of the kernel.

Naturally, we can treat the biases as node features similarly to how we handle them in an MLP. However, kernels cannot be treated as edge features in the same manner because their spatial dimensions differ from those of MLP weights. To address this problem, we propose to treat each weight scalar as an independent edge. Specifically, we construct edges between nodes of adjacent layers, $\mathbf{v}^{l-1}$ and $\mathbf{v}^l$, by a number of $c^{l-1} \times c^l \times h^l \times w^l$ edges, with each edge corresponding to a scalar in $\mathbf{W}^l$ and each pair of nodes is connected by $h^l \times w^l$ edges. To maintain consistency in the number of edges between each pair of nodes within the dynamic neural graph, we perform padding by adding additional zero edges between a pair of nodes to reach the maximum of $h^l \times w^l$.

## 5 Learning Invariant Latent Space on Dynamic Neural Graph

As discussed, modeling neural networks as dynamic graphs offers a natural alignment with the forward-pass computation. However, a key challenge remains: how should such dynamic neural graphs be effectively processed? Inspired by recent advances, notably the Temporal Graph Network (TGN) (Rossi et al.), we develop a tailored RNN-based encoder for dynamic neural graphs, which we term the *Dynamic Neural Graph Encoder* (DNG-Encoder). While this represents a foundational step, our approach opens the door to exploring more advanced architectures for dynamic neural graph processing.

**Message Passing.** Inspired by the process of multiplying activations by weights in a DNN, we use the linear complexity *conditional scaling* mechanism from FiLM (Perez et al., 2018) to define the Message Function for the DNG-Encoder. Notably, unlike the original FiLM, we do not include information about the target nodes and the shift module in our operation. This allows for a more faithful simulation of neural network inference, ensuring that weights are only multiplied by the activations of the previous layer.

---

[1]Due to page limit, we discuss the procedures to convert flattening layers and residual connections in Appendix F.1.

Below we present the Message Function for the case where there is only one edge between a pair of nodes (*i.e.,* for the dynamic neural graph converted by an MLP):

$$
\begin{aligned}
\mathbf{m}_i(t^l) &= \phi_m^{t^l}(\mathbf{s}_j(t^l-), \mathbf{e}_{ij}(t^l)) \\
&= \sum_{j \in \mathbf{N}_i} W_{m1}^{t^l} \mathbf{e}_{ij}(t^l) \odot W_{m2}^{t^l} \mathbf{s}_j(t^l-),
\end{aligned}
\tag{4}
$$

where $\mathbf{e}_{ij}(t^l)$ denotes the edge between the target node $\mathbf{v}_i$ and the source node $\mathbf{v}_j$ at time $t^l$. $\mathbf{s}_j(t^l-)$ represent the memories of $\mathbf{v}_j$ just before time $t^l$. $W_{m1}^{t^l}$ and $W_{m2}^{t^l}$ are two linear layers to perform linear transformations.

For a pair of nodes connected by multiple edges, say $N$ edges (i.e. for the dynamic neural graph converted by a CNN), we map the source node memory to $N$ heads. Each head interacts with one edge to generate multiple messages. Finally, we merge these messages through an MLP $\phi_h$:

$$
\mathbf{m}_i(t^l) = \sum_{j \in \mathbf{N}_i} \phi_h^{t^l} \Big( \mathrm{Concat}\big(head_{ij}^1(t^l), \ldots, head_{ij}^N(t^l)\big) \Big),
$$
$$
\text{where} \quad head_{ij}^n(t^l) = W_{m1}^{t^l} \mathbf{e}_{ij,n}(t^l) \odot W_n^{t^l} \mathbf{s}_j(t^l-). \tag{5}
$$

We illustrate the multi-head message passing function in Figure 3. It is worth noting that a similar operation, referred to as "multiple towers" (Gilmer et al., 2017), was proposed to address the computational challenges that arise when the dimensionality of node embeddings becomes excessively large. In contrast, our multi-head message function is primarily designed to ensure that a source node can transmit $N$ distinct messages to a target node through $N$ edges, thereby more effectively simulating the forward propagation process of a convolutional layer.

**Recurrent Memory Updating.** Recent works (Thost & Chen, 2021; Zhang et al., 2018) have demonstrated the effectiveness of using Gated Recurrent Units (GRUs) to capture complex dependencies during node representation updates. Inspired by these approaches, we similarly employ GRUs to update the memory of target nodes $\mathbf{v}_i^l$, enabling the capture of sequential dependencies between the layers of neural networks:

$$
\begin{aligned}
\mathbf{s}_i(t^l) &= \phi_u^{t^l}\big(\mathbf{m}_i(t^l), \mathbf{v}_i(t^l)\big) \\
&= \mathrm{LayerNorm}\Big( \mathrm{GRU}\big(\mathbf{m}_i(t^l), \mathbf{v}_i(t^l)\big) \Big).
\end{aligned}
\tag{6}
$$

Additionally, we incorporate layer normalization to stabilize the distribution of hidden state activations, improving training stability. Since the proposed DNG-Encoder processes the dynamic graph in a sequential manner, it differs from the MPNN used for static graphs (Kofinas et al., 2024). Therefore, we can avoid the introduction of the "inverse problem" discussed in Section 3.3. Due to page limit, for a comprehensive explanation of how we address these limitations, please refer to Appendix A.

It is worth mentioning that we do not update edge representations using the DNG-Encoder. For our dynamic neural graph framework, each edge is only utilized for message passing at a specific timestamp. Other than this timestamp, the edge is not included in the graph structure, meaning the same edge is not reused for multiple message-passing steps. For example, in Figure 2 (right), the edges in $\mathcal{G}_{t^1}$ are not present in the following processing timestamp. Therefore, updating edge features does not affect message passing process of our model. Additionally, the introduced dynamic neural graph allows us to simplify the temporal graph neural network framework by removing the message aggregator and embedding module, enhancing computational efficiency. The main reason is that each node in our dynamic neural network interacts only with the graph features at the current time, avoiding the memory staleness issue identified in Kazemi et al. (2020), which is typically managed by the embedding module. This property also eliminates the need for a memory aggregator to integrate messages received by a node from distinct events at various timestamps.

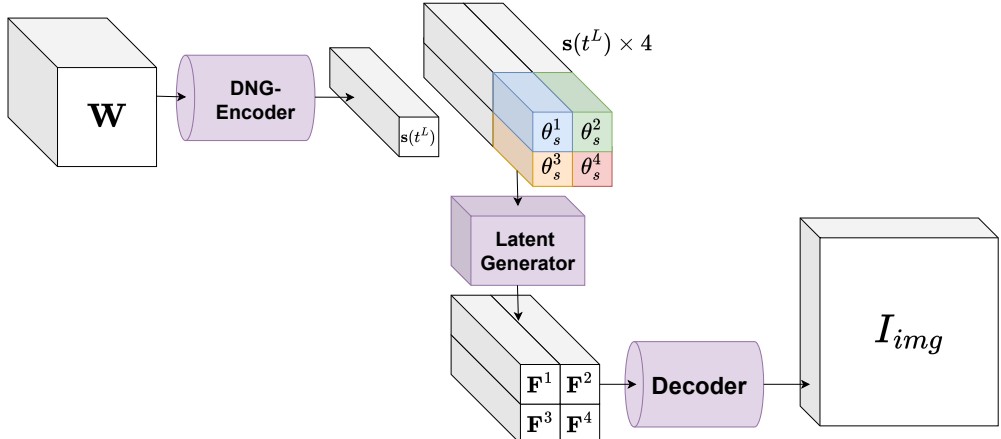

Figure 4: An overview of INR2JLS. Given the weights of an neural network, the DNG-Encoder processes the weights as dynamic neural graph to obtain the last recurrent memory. This memory state is combined with learnable spatial vectors in the Latent Generator to produce latent vectors, which are decoded into images using a transposed convolutional Decoder. The framework enables learning a joint latent space that links INR weights and image content without reconstructing the weights.

## 6 INR2JLS: Learning a Joint Latent Space from Data and Weights

Recent advancements in implicit neural representations (INRs) offer a novel approach to encoding continuous signals like images by mapping spatial coordinates directly to signal values. In this section, we treat INRs as specialized neural networks and develop an effective method to extract robust features for INR classification. Existing methods such as INR2VEC (De Luigi et al., 2023) and INR2ARRAY (Zhou et al., 2024b) have made great progress on the extracting representations from INRs for classification tasks. These approaches use encoder-decoder architectures to map INR weights into a latent space and reconstruct the INR weights. However, due to the high-dimensional nature of deep weights, this reconstruction process increases optimization difficulty, potentially leading to poor convergence and suboptimal latent spaces.

To this end, we introduce INR2JLS, a novel framework that supports randomly initialized neural network and provides a more informative latent space. Compared with the INR2VEC and INR2ARRAY, the main innovation of our INR2JLS is that we introduce a joint latent space between deep weights and the original data. To be specific, INR2JLS utilizes an encoder-decoder architecture to map INR weights to a latent representation capable of capturing both spatial and semantic information inherent in the original image. To achieve this, we do not decode the latent representations back to INR weights. Instead, we decode the latent representation to the original image represented by the input INR. We provide an overview of INR2JLS in Figure 4.

Our encoder, $\text{ENC}_\theta$, is built with the *DNG-Encoder* and a *Latent Generator* $\phi_g$. By transforming the input INR weights to dynamic neural graphs, we first employ DNG-Encoder to recurrently process the graph and obtain the last recurrent memory $\mathbf{s}(t^L)$. As discussed earlier, $\mathbf{s}(t^L)$ should inherit all the information contained in a complete forward pass of INR. However, we find that directly decoding an image from $\mathbf{s}(t^L)$ is extremely difficult. One potential reason is that $\mathbf{s}(t^L)$ may store very little spatial information of the original image. Inspired by the positional encoding technique, we introduce a set of learnable spatial vectors $\{\theta_s^1, ..., \theta_s^N\}$, where $N = h_s \times w_s$ denotes the dimension of latent representation. For generating a single latent vector, we have:

$$\mathbf{F}^n = \phi_g(\text{Concat}(\mathbf{s}(t^L), \theta_s^n)), \tag{7}$$

where $\mathbf{F}^n \in \mathbb{R}^d$ is the feature vector at the index $n$. We conduct Equation 7 over all the set $\{\theta_s^1, ..., \theta_s^N\}$. In this way, we have a set of feature vectors $\{\mathbf{F}^1, ..., \mathbf{F}^N\}$. Then, we reshape the set of feature vector to form a 3D feature map $\mathbf{F} \in \mathbb{R}^{h_s \times w_s \times d}$. Each latent vector in $\mathbf{F}$ corresponds to a spatial area in the original image.

Finally, we use transposed convolutional layers as a decoder $\text{DEC}_\theta$ to decode $\mathbf{F}$. The objective is to minimize the difference between the decoded outputs and the original images $I_{img}$. We use MSE as the loss function:

$$\mathcal{L}(\theta, \mathbf{W}) = \text{MSE}(\text{DEC}_\theta(\mathbf{F}), I_{img}), \tag{8}$$

$$\text{where} \quad \mathbf{F} = \text{ENC}_\theta(\mathbf{W}). \tag{9}$$

In image processing, implementing data augmentations is a common method to improve the generalization of trained networks. Current data augmentation techniques for deep space processing either encode augmented images into INRs (Kofinas et al., 2024; Zhou et al., 2024a) or directly modify the INR weights (Shamsian et al., 2024). With the proposed INR2JLS, we introduce a distinct augmentation method that can be easily implemented in our framework. Specifically, we generate different views of the original images using the decoder. By encouraging the INR2JLS to generate diverse views of the image $I_{img}$, the model can learn representations $\mathbf{F}_{aug}$ that are more robust and invariant to such transformations. Detailed discussion can be found in Appendix G.3.

## 7 Experiment for Comparing Static and Dynamic Neural Graph

To evaluate the ability of static and dynamic neural graph–based models to approximate the forward pass of an input neural network, we compare their performance in fitting layer-wise activations of multilayer perceptrons (MLPs) of varying depth.

**Data Generation.** We first generate 1,000 three-layer MLPs with randomly sampled weights and biases as training data, and another 500 structurally identical MLPs as testing data. For each MLP, we sample inputs and record the activation values at every layer, resulting in a dataset of 1,500 MLPs with corresponding activation trajectories.

**Model Setup.** Our goal is to use a message-passing neural network (MPNN) defined on either a static or a dynamic neural graph to produce node embeddings that fit the target activations. A smaller mismatch indicates a better approximation of the MLP's forward computation. To ensure a fair comparison, both graph variants adopt the same message function $\phi_m$ and node update function $\phi_u$ described in Section 3.2. Specifically, $\phi_m$ concatenates the source node features with the edge features and maps them through a two-layer MLP, while $\phi_u$ concatenates the aggregated incoming messages with the target node features and processes them through another two-layer MLP.

For the static neural graph, we implement the L-layer MPNN framework of (Kofinas et al., 2024), which uses an L-step message-passing process to mimic the forward propagation of an L-layer MLP, including their edge update mechanism. For the dynamic neural graph, we follow the construction in Section 4.1, where the graph evolves over timestamps to reflect the computation flow of the MLP, and node embeddings are updated layer by layer using the same $\phi_m$ and $\phi_u$.

**Results.** Figure 5 reports the MSE between predicted and ground-truth activations for MLPs with one, two, and three layers on the test set. The dynamic neural graph–based model maintains consistently low fitting error across all layers. In contrast, the static graph–based model performs similarly to the dynamic model only for the first layer; its accuracy degrades noticeably from the second layer onward, with the gap widening as depth increases.

These findings are consistent with our analysis in Section 3.3: the static neural graph framework in (Kofinas et al., 2024) is effective primarily at approximating the first forward-pass step of the input neural network, but struggles to capture computations in deeper layers. In comparison, our dynamic neural graph framework successfully models all stages of the forward pass, enabling accurate approximation of the computation flow.

## 8 Experiments on Downstream Applications

We here conduct a comprehensive evaluation of our method in accordance with (Navon et al., 2023; Kofinas et al., 2024; Zhou et al., 2024a). This evaluation involves a series of experiments across two tasks, each designed to utilize deep neural network weights as inputs. Specifically, the tasks are: (1) classifying INRs;

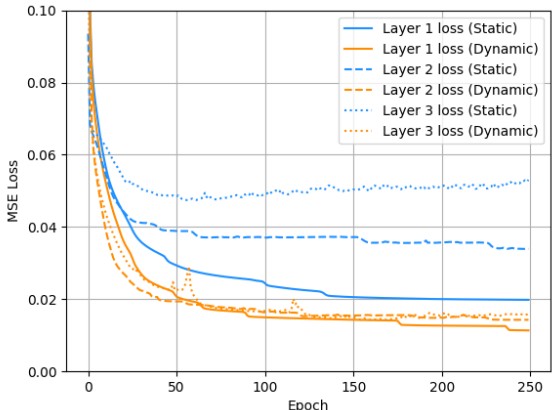

Figure 5: The Mean Squared Error (MSE) for fitting the activations of each layer of an MLP using the static neural graph-based model and the dynamic neural graph-based model, respectively.

|  | #Params | MNIST | FashionMNIST | CIFAR-10 | CIFAR-100 |
|---|---|---|---|---|---|
| NFN | $\sim$135M | $92.9_{\pm0.38}$ | $75.6_{\pm1.07}$ | $46.6_{\pm0.13}$ | $20.55_{\pm0.93}$ |
| INR2ARRAY (NFT) | $\sim$59M | $98.5_{\pm0.00}$ | $79.3_{\pm0.00}$ | $63.4_{\pm0.00}$ | $31.30_{\pm0.04}$ |
| NG-GNN | $\sim$6M | $97.3_{\pm0.02}$ | $86.53_{\pm0.58}$ | $55.11_{\pm1.43}$ | $26.50_{\pm1.32}$ |
| NG-T | $\sim$6M | $96.83_{\pm0.06}$ | $85.24_{\pm0.13}$ | $57.7_{\pm0.36}$ | $31.65_{\pm0.28}$ |
| INR2JLS (Ours) | $\sim$6M | $\mathbf{98.6}_{\pm0.01}$ | $\mathbf{90.6}_{\pm0.07}$ | $\mathbf{73.2}_{\pm0.28}$ | $\mathbf{42.4}_{\pm0.32}$ |

Table 1: INR classification accuracy (%) using 10-view INR augmentation across datasets. #Params is inference-time parameters. NG-GNN and NG-T use 64 probe features, scaled to match our model's inference parameters.

and (2) assessing the generalization capabilities of CNN classifiers by analyzing their weights. We compare the proposed method with several state-of-the-art methods, including NFN (Zhou et al., 2024a), NFT (Zhou et al., 2024b) and NG-GNN/NG-T (Kofinas et al., 2024). More implementation details are provided in Appendix G.

## 8.1 Classifying INRs with INR2JLS

**Experiment Setup.** There are mainly two steps in order to use our proposed framework for INR classification. First, we utilize the INR2JLS framework introduced in Section 6, which uses DNG-Encoder proposed in Section 5 along with a proposed augmentation strategy to generate a permutation-invariant implicit feature map, $F_{aug}$, that captures diverse semantic information from images via reconstruction. Our augmentation involves five transformations, including clockwise rotations of 90, 180 and 270 degrees, as well as horizontal and vertical flips. This increases the channels of $F_{aug}$ to six times of a single latent feature $\mathbf{F}$, represented as $\mathbf{F}_{aug} \in \mathbb{R}^{h_s \times w_s \times 6d}$.

Second, we keep the DNG-Encoder and the Latent Generator fixed and add additional classification CNN that takes $\mathbf{F}_{aug}$ as inputs. During training, given a set of INR weights and their corresponding label, the DNG-Encoder processes the INR weights to produce $\mathbf{F}_{aug}$. Then $\mathbf{F}_{aug}$ is used as input to the classification CNN, and we optimize the CNN using cross-entropy loss to learn a mapping from latent space to label space. It is worth emphasizing that during the training for classifying INRs, we do not directly optimize our encoder to encourage it extracting label-relevant features from input INRs. Instead, we employ a pre-trained encoder, which is learned in a self-supervised manner. In this case, the performance of classification can indicate the quality of the learned latent space.

**Comparison Results.** Table 1 presents a comparison between our method and several state-of-the-art approaches on the test sets of the four aforementioned datasets (MNIST, FashionMNIST, CIFAR-10, and CIFAR-100). Our method consistently exhibits superior performance compared to all other approaches.

| Model | CIFAR-10-GS | SVHN-GS | CNN Wild Park |
|---|---|---|---|
| NFN (HNP) | $0.934_{\pm 0.001}$ | $\mathbf{0.931_{\pm 0.005}}$ | - |
| NFN (NP) | $0.922_{\pm 0.001}$ | $0.856_{\pm 0.001}$ | - |
| NFT | $0.926_{\pm 0.001}$ | $0.858_{\pm 0.000}$ | - |
| NG-GNN | $0.930_{\pm 0.001}$ | $0.863_{\pm 0.002}$ | $0.8040_{\pm 0.0090}$ |
| NG-T | $0.935_{\pm 0.000}$ | $0.872_{\pm 0.001}$ | $0.8170_{\pm 0.0070}$ |
| DNG-Encoder (Ours) | $\mathbf{0.936_{\pm 0.000}}$ | $0.867_{\pm 0.002}$ | $\mathbf{0.8743_{\pm 0.0021}}$ |

Table 2: Kendall's rank correlation coefficient ($\tau$) for different models in predicting CNN generalization performance, where higher $\tau$ values indicate better prediction accuracy.

Notably, as the difficulty of the dataset increases, our method not only outperforms the best existing method but does so by an increasingly larger margin. Particularly noteworthy is the performance on the challenging CIFAR-10 and CIFAR-100 datasets, where our method surpasses other models by at least 9% and 10%, respectively. This indicates that our method effectively extracts representative features from deep weight spaces.

## 8.2 Predicting CNN Classifier Generalization

The above experiments evaluate the capability of our model to handle dynamic neural graphs built from the weight space of INRs. Here, we aim to evaluate the performance of our method on processing dynamic neural graphs built with CNN architecture. It is worth emphasizing that the objective of this experiment is to predict the test accuracy of a trained CNN classifier using its parameters. Following Zhou et al. (2024a), we conduct this experiment on the Small CNN Zoo (Unterthiner et al., 2020) dataset. This dataset contains thousands of CNNs trained on public image classification datasets. We follow Zhou et al. (2024a) to evaluate the CNNs trained on CIFAR-10-GS and SVHN-GS datasets. Furthermore, to ensure a rigorous evaluation of its robustness, we assessed our dynamic graph method on the challenging CNN Wild Park dataset (Kofinas et al., 2024). This benchmark is characterized by highly heterogeneous CNN architectures, featuring diverse configurations in terms of layers, channels, and skip connections.

We employ the DNG-Encoder to process the dynamic neural graph derived from the input CNNs. Subsequently, we add an MLP to map the recurrent memory $\mathbf{s}(t^L)$ of the last graph layer at the final timestamp to the predicted test accuracy of the CNN. We use binary cross-entropy loss in the training. For models tested on the CNN Wild Park dataset, we ensure a strictly fair comparison by aligning our model's parameter count ($\sim 0.4$M) with the NG baselines.

Table 2 shows the test performance of different models on the CIFAR-10-GS, SVHN-GS, and CNN Wild Park datasets using the rank correlation $\tau$ (Kendall, 1938) as the metric. It can be observed that our model outperforms all other methods on the CIFAR-10-GS dataset. Regarding the performance on the SVHN-GS dataset, we acknowledge that NFN(HNP) exhibits strong performance in that specific setting. However, a critical limitation of NFN(HNP) (and similar weight-sharing methods such as NFN(NP) and NFT) is that they are structurally restricted to homogeneous network architectures (i.e., identical layer counts and widths). They fundamentally cannot process the highly heterogeneous architectures found in CNN Wild Park or real-world model zoos. In contrast, our dynamic graph framework is architecture-agnostic and effortlessly generalizes across diverse network topologies, providing a much broader and more practical utility. This is strongly evidenced by our DNG-Encoder significantly outperforming the static graph baselines (NG-GNN and NG-T) on the heterogeneous CNN Wild Park dataset.

## 9 Further Empirical Analysis

To further demonstrate the effectiveness of each module/components in the proposed INR2JLS, we present a comprehensive analysis below.

| Study Type | Method | MNIST | FashionMNIST | CIFAR-10 | CIFAR-100 |
|---|---|---|---|---|---|
| Recon. Study | INR2JLS (Ours) | $\mathbf{98.6}_{\pm 0.01}$ | $\mathbf{90.6}_{\pm 0.07}$ | $\mathbf{73.2}_{\pm 0.28}$ | $\mathbf{42.4}_{\pm 0.32}$ |
| | INR-INR | $98.6_{\pm 0.08}$ | $88.3_{\pm 0.04}$ | $56.3_{\pm 0.25}$ | $30.6_{\pm 0.16}$ |
| Modules' Study | DNG-Encoder | $96.6_{\pm 0.09}$ | $78.4_{\pm 0.61}$ | $54.0_{\pm 0.07}$ | $25.7_{\pm 0.12}$ |
| | INR2JLS w/o Latent Generator | $98.4_{\pm 0.08}$ | $88.9_{\pm 0.28}$ | $54.5_{\pm 0.51}$ | $28.1_{\pm 0.43}$ |
| | INR2JLS (Ours) | $\mathbf{98.6}_{\pm 0.01}$ | $\mathbf{90.6}_{\pm 0.07}$ | $\mathbf{73.2}_{\pm 0.28}$ | $\mathbf{42.4}_{\pm 0.32}$ |

Table 3: Top: Ablation study on image reconstruction in INR2JLS. Bottom: Ablation study on the importance of different modules (DNG-Encoder, Latent Generator) in INR2JLS. Results are shown in classification accuracy (%).

| | MNIST | FashionMNIST | CIFAR-10 | CIFAR-100 |
|---|---|---|---|---|
| No Augmentation | $98.5_{\pm 0.00}$ | $89.5_{\pm 0.07}$ | $66.4_{\pm 0.19}$ | $32.9_{\pm 0.31}$ |
| Adding Noise | $98.4_{\pm 0.01}$ | $89.5_{\pm 0.06}$ | $67.3_{\pm 0.38}$ | $33.0_{\pm 0.24}$ |
| Rotation & Flip | $\mathbf{98.6}_{\pm 0.01}$ | $\mathbf{90.6}_{\pm 0.07}$ | $\mathbf{73.2}_{\pm 0.28}$ | $\mathbf{42.4}_{\pm 0.32}$ |

Table 4: Test accuracy (%) of INR classification using INR2JLS, with/without data augmentation.

### 9.1 Analysis of the Data Augmentation.

We conduct an analysis on the effectiveness of the augmentation strategy introduced in Section 6. By comparing the third and fifth row in Table 4, it can be found that the rotation and flip augmentation can help improve the classification accuracy significantly. For example, on complex datasets such as CIFAR-10 / CIFAR-100, using rotation and flip augmentations can improve over baseline by around 7% and 9%.

### 9.2 The Importance of Image Reconstruction in the INR2JLS.

First, we conduct an experiment to compare the performance of two frameworks, our proposed image reconstruction framework (INR2JLS), and the INR-weight reconstruction framework (INR-INR), on the INR classification task. For the INR-INR framework, we employ the DNG-Encoder but use two MLPs to map the node memory, generated by the encoder, to the weights and biases of an INR. Subsequently, we adopt a methodology same as the NFT (Zhou et al., 2024b) to compute the loss between the image obtained by the reconstructed INR and the original image. Finally, an MLP is employed to classify the INR based on the node memory output from the pretrained encoder. Table 3 (Top) shows the classification performance of the two frameworks on INR datasets. Our INR2JLS outperforms INR-INR, highlighting the significance of learning a joint space between INR and the original images.

### 9.3 Ablation Study of Key Components in INR2JLS.

We here conduct ablation experiments to assess the individual contributions of the two modules (Decoder and Latent Generator) within the INR2JLS framework towards enhancing performance in the INR classification task. In the first ablation experiment, we remove the decoder from INR2JLS and directly add an MLP classifier on the output of the encoder (node memory). In the second ablation experiment, we remove the Latent Generator from the INR2JLS framework, and employ an MLP decoder to directly map the node memory obtained from the DNG-Encoder to images for the reconstruction task. We then utilize another MLP to classify the node memory generated by the pretrained DNG-Encoder. Table 3 (Bottom) shows the experimental results. Our INR2JLS significantly outperforms its two variants with component removal, further demonstrating the necessity of the proposed Decoder and Latent Generator.

### 9.4 Efficiency analysis.

Table 5 compares running time, memory usage, and computational complexity for processing a single INR. INR2JLS is significantly faster than other methods, with slightly higher memory usage than NG-GNN but much lower than NFN and NFT, and has the lowest computational complexity overall. We believe these advantages make INR2JLS a efficient choice for processing neural network weights.

|  | Run Time (s) | Memory (MB) | **Comp. Cost** (GFLOPs) |
|---|---|---|---|
| NFN | $0.0082_{\pm 0.00009}$ | 273.08 | 2.58 |
| NFT | $0.0527_{\pm 0.00170}$ | 241.15 | 10.60 |
| NG-GNN | $0.0124_{\pm 0.00070}$ | **27.40** | 2.13 |
| NG-T | $0.0092_{\pm 0.00041}$ | 29.77 | 14.82 |
| INR2JLS (Ours) | $\mathbf{0.0047}_{\pm 0.00018}$ | 29.17 | **1.31** |

Table 5: Inference efficiency comparison for INR classification on the MNIST INR dataset, measured by the inference time of a single INR for each method.

### 9.5 Analysis of Positional Encoding in INR2JLS

Our method uses an RNN-based GNN to process network weights sequentially, naturally capturing layer order without requiring explicit positional encoding. To verify this design choice, we add learnable positional embeddings to node features at different layers and evaluate their effect (Table 6). The results show no notable performance improvement across the four INR classification datasets, indicating that positional encoding is unnecessary for our framework.

| Method Variant | MNIST | FashionMNIST | CIFAR-10 | CIFAR-100 |
|---|---|---|---|---|
| **Effect of Positional Encoding** | | | | |
| INR2JLS with positional encoding | 98.6±0.02 | 89.9±0.09 | 73.5±0.04 | 42.7±0.16 |
| INR2JLS (Ours) | 98.6±0.01 | 90.6±0.07 | 73.2±0.28 | 42.4±0.32 |
| **Effect of Non-linearity Embeddings** | | | | |
| INR2JLS add non-linearity embeddings | 98.4±0.16 | 90.4±0.01 | 73.2±0.12 | 42.6±0.04 |
| INR2JLS (Ours) | 98.6±0.01 | 90.6±0.07 | 73.2±0.28 | 42.4±0.32 |

Table 6: Ablation study on INR2JLS. We analyze (1) the effect of positional encoding and (2) the effect of non-linearity embeddings. Results are reported as classification test accuracy (%) across four datasets.

### 9.6 Analysis of Non-linearity Embeddings in INR2JLS

In Kofinas et al. (2024), non-linearities are explicitly encoded as node features because their update function shares parameters across all layers, making such embeddings necessary to distinguish activation types. In contrast, as shown in Equation 4, our framework applies separate GNN layers to each timestamp. Given the expressivity of GNNs on neural networks (Appendix B of Kofinas et al. (2024)), each layer can directly learn its own activation behavior, eliminating the need for explicit non-linearity embeddings. To validate this, we augment our model with learnable non-linearity embeddings following Kofinas et al. (2024). As reported in Table 6, their inclusion yields no meaningful performance change across all four INR classification datasets, confirming that explicit non-linearity encoding is unnecessary in our framework.

## 10 Conclusion and Limitation

In this paper, we have introduced a novel method to model neural network weights as dynamic graphs. To process these structures, we propose the Dynamic Neural Graph Encoder (DNG-Encoder) to capture the temporal dynamics intrinsic to neural network inference, thereby maintaining the sequential flow of data through layers. We further enhance the model's utility via INR2JLS, which maps INR weights into a joint latent space to provide a more informative and robust representation for downstream tasks. Extensive experiments demonstrate the effectiveness of our approach across various benchmarks.

While our framework significantly advances the analysis of deep weight spaces, it is important to explicitly define the applicability boundaries of different weight-space representations. Our dynamic graphs excel in representation learning and discriminative tasks involving deep networks (e.g., INR classification, generalization prediction), where capturing the layer-by-layer forward pass resolves the inverse problem and preserves deep functional semantics. However, we acknowledge that simpler static or tokenized representations may

suffice in certain scenarios. For weight editing and generative tasks (e.g., INR style editing), static graphs with explicit edge-update mechanisms or tokenized models can be particularly suitable. While our current DNG architecture is primarily designed for node state evolution, extending it with dedicated mechanisms for generative edge reconstruction is an interesting direction for future work.

Beyond architectural applicability, we also acknowledge the potential dual-use implications of this emerging field. Capabilities that allow for inferring detailed properties directly from weights could be leveraged for unauthorized model attribution or the reverse-engineering of proprietary checkpoints. Consequently, we encourage the community to concurrently explore defensive mechanisms, such as weight obfuscation and robust watermarking, to safeguard neural architectures. Finally, despite our advancements, our method's performance on INR classification still lags behind that of CNNs on analogous image-space tasks. Bridging this performance gap and developing more powerful temporal GNN variants remain promising directions for future improvement.

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

## A How Does the DNG-Encoder Exhibit the Expressiveness of Neural Networks?

Using the DNG-Encoder to update nodes in dynamic neural graph ideally simulates the sequential updating pattern of neural networks. This approach effectively avoids the inverse problem typically encountered in static neural graphs as discussed in Section 3.3, so as to better exhibit the expressiveness of neural networks. Below, we discuss how the DNG-Encoder model better represents the expressiveness of MLPs compared to the static neural graph-based model.

An MLP $\mathbf{M}$ can be transformed into a dynamic neural graph $\mathcal{G}_T$ or a static neural graph $\mathcal{G}_s$. In $\mathcal{G}_T$, we update node representations asynchronously at each layer by modifying the graph structure at each timestamp to align with the forward pass process of $\mathbf{M}$. The graph structure under each timestamp of $\mathcal{G}_T$ is a layer-by-layer snapshot taken from $\mathcal{G}_s$. It contains the neurons involved in the computation of the neural network in each forward pass step and simulates the topology of these neurons. For example, the graph state $\mathcal{G}_{t^l}$ $(0 < l \leq L)$ of $\mathcal{G}_T$ only contains nodes at the $l$-th layer and the $(l-1)$-th layer of $\mathcal{G}_s$ and edges at the $l$-th layer of $\mathcal{G}_s$. The initial representations of all nodes and edges present in $\mathcal{G}_T$ over the time span $T = [t^0 : t^L]$ can be defined as $\{\mathbf{v}(t^0), \mathbf{v}(t^1), ..., \mathbf{v}(t^L)\}$ and $\{\mathbf{e}(t^1), ..., \mathbf{e}(t^L)\}$. Similar to $\mathcal{G}_s$, these nodes and edges represent the biases and weights of each layer in $\mathbf{M}$. According to DNG-Encoder defined in Section 5, the memory of target nodes $\mathbf{s}_i(t^l)$ at timestamp $t^l$ can be obtained by using the following equation:

$$\mathbf{s}_i(t^l) = \phi_u(\sum_{j \in N_i} \phi_m^{t^l}\left(\mathbf{e}_{ij}\left(t^l\right), \mathbf{s}_j\left(t^l-\right)\right), \mathbf{v}_i(t^l)), \tag{10}$$

where $\phi_m^{t^l}$ is the message function at $t^l$, corresponding to the Message Function in the DNG-Encoder. $\phi_u$ is the node update function shared for all timestamps, corresponding to the GRU module in DNG-Encoder. For example, at the first timestamp $t^1$, $\mathbf{e}_{ij}(t^1)$ is an initial representation of an edge newly added at $t^1$, corresponding to $\mathbf{W}_{ij}^1$ in Equation 1, while $\mathbf{s}_j(t^1-)$ corresponds to the input $\mathbf{x}_j$ in Equation 1. From Equation 4, we know that $\phi_m^{t^1}$ can approximate the multiplication operation between two given inputs. Thus, given $\mathbf{e}_{ij}(t^1)$ and $\mathbf{s}_j(t^1-)$, its output can represent $\mathbf{W}_{ij}^1 \mathbf{x}_j$ in Equation 1. $\mathbf{v}_i(t^1)$ is the initial representation of a newly added node at $t^1$, corresponding to $\mathbf{b}_i^1$ in Equation 1. From Equation 6, given the aggregation of the outputs of $\phi_m^{t^1}$ and $\mathbf{v}_i(t^1)$, $\phi_u$ can easily approximate the computation of adding $\mathbf{b}_i^1$ to $\sum_j \mathbf{W}_{ij}^1 \mathbf{x}_j$ and then applying an activation function. In this way, we say that $\mathbf{s}_i(t^1)$ can directly represent $\mathbf{a}_i^1$.

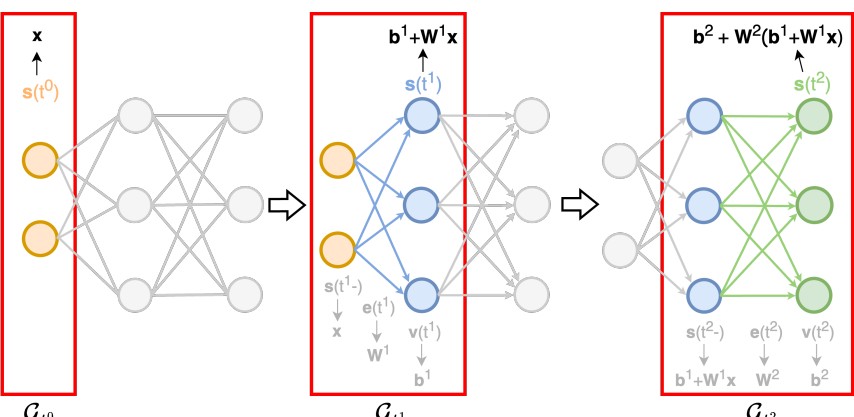

Figure 6: An illustration of how dynamic neural graphs and the DNG-Encoder simulate the forward pass of a neural network while addressing the limitations present in static neural graphs.

According to the discussion in Section 3.2 and 3.3, it is evident that the static neural graph-based model can also approximate the first forward pass step of $\mathbf{M}$ easily using one MPNN layer. However, it faces challenges in approximating subsequent forward pass steps due to the emergence of the "inverse problem". The following demonstrates how the DNG-Encoder avoids the inverse problem during computation, thereby

enabling it to easily approximate the forward pass process for all steps of $\mathbf{M}$. We here define the process of $\mathbf{M}$ to obtain the $i$-th activation $\mathbf{a}_i^l$ at the $l$-th layer as follows:

$$\mathbf{a}_i^l = \sigma(\sum_j \mathbf{W}_{ij}^l \mathbf{a}_j^{l-1} + \mathbf{b}_i^l). \tag{11}$$

At any timestamp $t^l$, $\mathbf{v}_i(t^l)$ in Equation 10 is the initial representation of the newly added node, corresponding to $\mathbf{b}_i^l$ in Equation 11. Smilarily, $\mathbf{e}_{ij}(t^l)$ in Equation 10 is the initial representation of the newly added edge, corresponding to $\mathbf{W}_{ij}^l$ in Equation 11. Besides, $\mathbf{s}_j(t^l-)$ in Equation 10 corresponds to $\mathbf{a}_j^{l-1}$.

Following the expressivity in Kofinas et al. (2024), this suggests that the message functions $\phi_m^{t^l}, \ldots, \phi_m^{t^L}$, which have the same structure at all timestamps, along with the shared update function $\phi_u$, can accurately model all forward pass steps of $\mathbf{M}$. Since the inputs to the message/update functions are simple and do not contain extra complex terms, the approximation is straightforward, eliminating the risk of inverse problems and allowing $\mathbf{s}_i(t^l)$ to directly represent $\mathbf{a}_i^l$. Therefore, the proposed DNG-Encoder can maximally approximate the forward pass of $\mathbf{M}$. Figure 6 intuitively illustrates how dynamic neural graphs and the DNG-Encoder eliminate the risk of inverse problems and effectively simulate the forward pass of $\mathbf{M}$.

## B  Disentangling the Contribution of the Dynamic Graph Architecture

We conduct an additional experiment on the INR classification task to explicitly evaluate and disentangle the contribution of the dynamic graph itself from the broader INR2JLS framework. This experiment provides a direct, intuitive task-level comparison between graph architectures.

To ensure strict fairness and to emphasize the intrinsic expressiveness of the graph frameworks, we compare our DNG-Encoder directly against the static graph baselines, NG-GNN and NG-T (Kofinas et al., 2024). In all cases, we use *only* the graph encoders followed by a standard MLP classification head, entirely removing the joint latent-space supervision and data augmentations of the full INR2JLS pipeline. Crucially, we also remove the probe features from the NG-GNN and NG-T models. Probe features are intermediate outputs generated by feeding specific inputs during the forward pass; because they explicitly leak dynamic forward-pass information into the static graph, they act as a shortcut that bypasses the static structure's inherent limitations. Excluding them is therefore essential to observe the true capabilities of the static graph versus dynamic graph structures themselves.

Table 7 presents the classification performance of the three models on the MNIST, FashionMNIST, and CIFAR-10 INR datasets. The results reveal that simply upgrading the architecture from a static graph (NG-T) to our dynamic graph (DNG-Encoder) yields a massive performance leap (e.g., +9.31% on CIFAR-10). This empirical evidence confirms that dynamic graphs provide a fundamentally superior and more expressive architectural backbone for capturing the semantics of neural networks compared to static graphs.

Table 7: Test accuracy (%) for the INR classification task comparing pure graph encoders without augmentations or joint latent-space supervision. Probe features are excluded from the static NG baselines to isolate the structural expressiveness of the graphs. Our dynamic DNG-Encoder consistently and significantly outperforms the static graph-based classifiers across all three datasets.

| Method | #Params | MNIST | FashionMNIST | CIFAR-10 |
|---|---|---|---|---|
| NG-GNN (Static) | ∼0.3M | 79.60±1.30 | 71.10±0.42 | 43.94±0.06 |
| NG-T (Static) | ∼0.4M | 83.43±0.12 | 72.13±0.51 | 44.69±0.03 |
| **DNG-Encoder (Ours)** | ∼0.4M | **96.60**±0.09 | **78.40**±0.61 | **54.00**±0.07 |

# C  Dynamic Neural Graph Symmetry

## C.1  Background and Definitions

### C.1.1  Neuron Permutation Symmetry in Neural Networks

Consider an $L$-layer Multilayer Perceptron (MLP) $\mathbf{M}$ with weight matrices $\{\mathbf{W}^l\}_{l=1}^L$ and biases $\{\mathbf{b}^l\}_{l=1}^L$. The network computes activations as:

$$\mathbf{h}^l = \sigma\left(\mathbf{W}^l \mathbf{h}^{l-1} + \mathbf{b}^l\right), \quad \text{for } l = 1, 2, \dots, L, \tag{12}$$

where $\mathbf{h}^0$ is the input and $\sigma$ is an activation function.

**Neuron Permutation Symmetry:** Permuting the neurons within a hidden layer $l$ and appropriately adjusting the corresponding rows and columns of the weight matrices and biases leaves the function represented by the network unchanged. Formally, for any permutation $\pi^l$ of the neurons in layer $l$, there exists a transformed network $\tilde{\mathbf{M}}$ such that:

$$\tilde{\mathbf{W}}^l = \mathbf{P}^{\pi^l} \mathbf{W}^l \left(\mathbf{P}^{\pi^{l-1}}\right)^\top, \quad \tilde{\mathbf{b}}^l = \mathbf{P}^{\pi^l} \mathbf{b}^l, \tag{13}$$

where $\mathbf{P}^{\pi^l}$ is the permutation matrix corresponding to $\pi^l$.

## C.2  Objective

Our goal is to prove that the dynamic neural graph $\mathcal{G}_T$ is equivariant to neuron permutations in the MLP $\mathbf{M}$. That is, permuting the neurons in any layer $l$ corresponds to permuting the nodes $\mathbf{v}^l$ in $\mathcal{G}_T$, and the graph update operations $\mathbf{O}_{t^l}$ are consistent under such permutations.

## C.3  Proof of Equivariance

### C.3.1  Defining Permutations in Neural Networks and Graphs

Let $\pi^l$ be a permutation of the neurons in layer $l$ of the MLP, and let $\mathbf{P}^{\pi^l}$ be the corresponding permutation matrix. The permutation acts on the weights and biases as in Equation 13.

In the dynamic neural graph, we adopt the convention that the permutation $\pi^l$ relocates node $i$ to position $\pi^l(i)$, i.e., $\tilde{\mathbf{v}}^l_{\pi^l(i)} = \mathbf{v}^l_i$, which is equivalent to $\tilde{\mathbf{v}}^l = \mathbf{P}^{\pi^l} \mathbf{v}^l$ with $(\mathbf{P}^{\pi^l})_{ab} = \mathbf{1}[a = \pi^l(b)]$. Under this convention, $\pi^l$ acts on the nodes $\mathbf{v}^l$ and edges $\mathbf{e}^l$ as follows:

- **Nodes:** The permuted nodes satisfy $\tilde{\mathbf{v}}^l = \mathbf{P}^{\pi^l} \mathbf{v}^l$.

- **Edges:** Each edge from node $j$ in $\mathbf{v}^{l-1}$ to node $i$ in $\mathbf{v}^l$ becomes an edge from node $\pi^{l-1}(j)$ to node $\pi^l(i)$ after permutation.

### C.3.2  Node Permutations Correspond to Neuron Permutations

Since each node $\mathbf{v}^l_i$ corresponds to neuron $i$ in layer $l$ of the MLP, permuting the neurons directly corresponds to permuting the nodes:

$$\tilde{\mathbf{v}}^l_{\pi^l(i)} = \mathbf{v}^l_i. \tag{14}$$

### C.3.3 Edge Adjustments Under Permutation

Edges $\mathbf{e}^l$ represent the connections (weights) between nodes in $\mathbf{v}^{l-1}$ and $\mathbf{v}^l$. The adjacency matrix $\mathbf{A}^l$ corresponding to edges $\mathbf{e}^l$ is related to the weight matrix $\mathbf{W}^l$.

Under permutations $\pi^{l-1}$ and $\pi^l$, the adjacency matrix transforms as:

$$\tilde{\mathbf{A}}^l = \mathbf{P}^{\pi^l} \mathbf{A}^l \left( \mathbf{P}^{\pi^{l-1}} \right)^\top. \tag{15}$$

This ensures that the structure of the graph remains consistent with the permuted network.

### C.3.4 Equivariance of Graph Update Operations

The graph update operations $\mathbf{O}_{t^l}$ are defined independently of node identities and depend only on the layer structure. Therefore, they are consistent under permutations:

$$\tilde{\mathbf{O}}_{t^l} = \mathbf{O}_{t^l}. \tag{16}$$

Applying the joint permutation $(\pi^{l-1}, \pi^l)$ to $\mathcal{G}_{t^l}$ after the graph update operations yields:

$$\tilde{\mathcal{G}}_{t^l} = (\pi^{l-1}, \pi^l) \cdot \mathcal{G}_{t^l}, \tag{17}$$

where $(\pi^{l-1}, \pi^l)$ permutes nodes in $\mathbf{v}^{l-1}$ and $\mathbf{v}^l$ via $\pi^{l-1}$ and $\pi^l$ respectively, and updates the edges accordingly.

### C.3.5 Inductive Proof Over Layers

We use mathematical induction over the layers $l$ to show that the graph remains equivariant under permutations.

**Base Case** $(l = 1)$  At $t^1$:

- The graph $\mathcal{G}_{t^1}$ consists of input nodes $\mathbf{v}^0$ and nodes $\mathbf{v}^1$.
- Permuting $\mathbf{v}^1$ corresponds to permuting neurons in layer 1.
- The update operations $\mathbf{O}_{t^1}$ are equivariant under $\pi^1$.

**Inductive Step**  Assume $\mathcal{G}_{t^{l-1}}$ is equivariant under permutations up to layer $l - 1$. At $t^l$:

- Applying $\mathbf{O}_{t^l}$ to $\mathcal{G}_{t^{l-1}}$ adds nodes $\mathbf{v}^l$ and edges $\mathbf{e}^l$.
- Under permutation $\pi^l$, nodes and edges are permuted as per Equations 14 and 15.
- Thus, $\mathcal{G}_{t^l}$ remains equivariant under the combined permutations $\pi^{l-1}$ and $\pi^l$.

By induction, $\mathcal{G}_T$ is equivariant under neuron permutations at each layer.

## D  Equivariance of the DNG-Encoder on Dynamic Graphs

In this section, we prove that our proposed DNG-Encoder, when applied to dynamic graphs, is *equivariant* under node permutations. This property ensures that if the nodes of the input graph are permuted, the output will be permuted in the same way, maintaining consistency regardless of the node ordering.

### D.1 Definition of Equivariance

A function $F$ operating on graphs is said to be **equivariant** to node permutations if, for any permutation $\pi$ and input graph $\mathcal{G}$, the following holds:

$$F(\pi \cdot \mathcal{G}) = \pi \cdot F(\mathcal{G}), \tag{18}$$

where $\pi \cdot \mathcal{G}$ denotes the graph obtained by permuting the nodes of $\mathcal{G}$ according to $\pi$, and similarly for $\pi \cdot F(\mathcal{G})$.

### D.2 Equivariance of the Message Passing Function

Our message passing function is defined differently for single-edge and multi-edge cases.

**Single-Edge Case** For the case where there is only one edge between a pair of nodes (i.e., for the dynamic neural graph converted from an MLP), the message function is defined in Equation 4 as:

$$\mathbf{m}_i(t^l) = \sum_{j \in \mathcal{N}_i} \phi_m^{t^l}\big(\mathbf{s}_j(t^{l-}), \mathbf{e}_{ij}(t^l)\big) = \sum_{j \in \mathcal{N}_i} W_{m1}^{t^l}\mathbf{e}_{ij}(t^l) \odot W_{m2}^{t^l}\mathbf{s}_j(t^{l-}), \tag{19}$$

**Proof of Equivariance:**

Let $\pi$ be a permutation of the node indices. Recall that, following Equation 19, $\mathbf{e}_{ij}(t^l)$ denotes the edge from source node $j$ to target node $i$. Under permutation $\pi$:

- Node $i$ is relocated to position $\pi(i)$.

- The neighbor set $\mathcal{N}_i$ becomes $\mathcal{N}_{\pi(i)} = \{\pi(j) \mid j \in \mathcal{N}_i\}$.

- The edge from $j$ to $i$ becomes the edge from $\pi(j)$ to $\pi(i)$.

- Node states and edge features are permuted accordingly:

$$\mathbf{s}'_{\pi(j)}(t^{l-}) = \mathbf{s}_j(t^{l-}), \quad \mathbf{e}'_{\pi(i)\pi(j)}(t^l) = \mathbf{e}_{ij}(t^l). \tag{20}$$

The message for node $\pi(i)$ after permutation is:

$$\begin{aligned}
\mathbf{m}'_{\pi(i)}(t^l) &= \sum_{k \in \mathcal{N}_{\pi(i)}} W_{m1}^{t^l}\mathbf{e}'_{\pi(i)k}(t^l) \odot W_{m2}^{t^l}\mathbf{s}'_k(t^{l-}) \\
&= \sum_{j \in \mathcal{N}_i} W_{m1}^{t^l}\mathbf{e}'_{\pi(i)\pi(j)}(t^l) \odot W_{m2}^{t^l}\mathbf{s}'_{\pi(j)}(t^{l-}) \\
&= \sum_{j \in \mathcal{N}_i} W_{m1}^{t^l}\mathbf{e}_{ij}(t^l) \odot W_{m2}^{t^l}\mathbf{s}_j(t^{l-}) \\
&= \mathbf{m}_i(t^l).
\end{aligned} \tag{21}$$

Therefore, we have:

$$\mathbf{m}'_{\pi(i)}(t^l) = \mathbf{m}_i(t^l), \tag{22}$$

which shows that the message passing function is equivariant under node permutations in the single-edge case.

**Multi-Edge Case** For the case where there are multiple edges between a pair of nodes (i.e., for the dynamic neural graph converted from a CNN), the message function is:

$$\mathbf{m}_i(t^l) = \sum_{j \in \mathcal{N}_i} \phi_h^{t^l}\left(\mathrm{Concat}\left(\mathrm{head}_{ij}^1(t^l), \ldots, \mathrm{head}_{ij}^N(t^l)\right)\right), \tag{23}$$

where each head is defined as:

$$\text{head}_{ij}^n(t^l) = W_{m1}^{t^l}\mathbf{e}_{ij,n}(t^l) \odot W_n^{t^l}\mathbf{s}_j(t^{l-}). \tag{24}$$

**Proof of Equivariance:**

Under permutation $\pi$, similar reasoning applies:

- Edge features are permuted: $\mathbf{e}'_{\pi(i)\pi(j),n}(t^l) = \mathbf{e}_{ij,n}(t^l)$.

- Node states are permuted: $\mathbf{s}'_{\pi(j)}(t^{l-}) = \mathbf{s}_j(t^{l-})$.

The message for node $\pi(i)$ is:

$$
\begin{aligned}
\mathbf{m}'_{\pi(i)}(t^l) &= \sum_{k \in \mathcal{N}_{\pi(i)}} \phi_h^{t^l}\left(\text{Concat}\left(\text{head}_{\pi(i)k}^1(t^l), \ldots, \text{head}_{\pi(i)k}^N(t^l)\right)\right) \\
&= \sum_{j \in \mathcal{N}_i} \phi_h^{t^l}\left(\text{Concat}\left(\text{head}_{\pi(i)\pi(j)}^1(t^l), \ldots, \text{head}_{\pi(i)\pi(j)}^N(t^l)\right)\right) \\
&= \sum_{j \in \mathcal{N}_i} \phi_h^{t^l}\left(\text{Concat}\left(\text{head}_{ij}^1(t^l), \ldots, \text{head}_{ij}^N(t^l)\right)\right) \\
&= \mathbf{m}_i(t^l).
\end{aligned}
\tag{25}
$$

Thus, the message passing function remains equivariant under node permutations in the multi-edge case as well.

### D.3 Equivariance of the Recurrent Memory Updating

The recurrent memory update function is defined as:

$$\mathbf{s}_i(t^l) = \phi_u^{t^l}(\mathbf{m}_i(t^l), \mathbf{v}_i(t^l)) = \text{LayerNorm}\big(\text{GRU}(\mathbf{m}_i(t^l), \mathbf{v}_i(t^l))\big), \tag{26}$$

where $\mathbf{v}_i(t^l)$ is the feature of node $i$ at time $t^l$. Both GRU and LayerNorm are applied per node with parameters shared across all nodes, so they preserve permutation equivariance.

Under permutation $\pi$, node features are permuted:

$$\mathbf{v}'_{\pi(i)}(t^l) = \mathbf{v}_i(t^l). \tag{27}$$

Messages are permuted:

$$\mathbf{m}'_{\pi(i)}(t^l) = \mathbf{m}_i(t^l). \tag{28}$$

Therefore, the updated memory state for node $\pi(i)$ is:

$$
\begin{aligned}
\mathbf{s}'_{\pi(i)}(t^l) &= \phi_u^{t^l}(\mathbf{m}'_{\pi(i)}(t^l), \mathbf{v}'_{\pi(i)}(t^l)) \\
&= \text{LayerNorm}\big(\text{GRU}(\mathbf{m}_i(t^l), \mathbf{v}_i(t^l))\big) \\
&= \mathbf{s}_i(t^l).
\end{aligned}
\tag{29}
$$

This shows that the memory update function is equivariant under node permutations.

### D.4 Equivariance of Graph Update Events

Our dynamic graph evolves through graph update events defined at each time $t^l$, as specified in Equation 3. These events include node addition $(+V)$, edge addition $(+E)$, node deletion $(-V)$, and edge deletion $(-E)$.

Under permutation $\pi$:

- Added nodes $\mathbf{v}^l$ become $\mathbf{v}^{l\prime} = \{\pi(i) \mid \mathbf{v}_i^l \in \mathbf{v}^l\}$.

- Added edges $\mathbf{e}^l$ become $\mathbf{e}^{l\prime} = \{(\pi(i), \pi(j)) \mid (\mathbf{v}_i^{l-1}, \mathbf{v}_j^l) \in \mathbf{e}^l\}$.

- Deleted nodes and edges are permuted similarly.

Since the graph update operations are applied consistently to the permuted nodes and edges, the sequence of graph updates remains equivariant under node permutations.

### D.5 Conclusion

By demonstrating that each component of our DNG-Encoder is equivariant under node permutations, we establish that the entire model maintains equivariance when applied to dynamic graphs. This property ensures that the model's outputs are consistent regardless of the node ordering, capturing the intrinsic structure of the graph without being influenced by arbitrary node labels.

## E Analysis on Computational Cost

Since our method does not require a decoder for inference, we focus this analysis on providing theoretical results for the computational costs during the inference within the encoder.

To determine the time complexity of applying an $L$-layer Message Passing Neural Network (MPNN) on a graph, we consider the following characteristics:

- **Number of nodes per MLP layer:** $n$. For simplicity, we assume each MLP layer has $n$ neurons.

- **Dimension of node/edge features:** $d$. For simplicity, we assume the dimensions of edge and node features are the same.

- **Number of MLP layers:** $L$.

### E.1 Time Complexity of Our Method

1. **Message Computation (Equation 4):**
   - Per edge computation: $O(d^2)$ (including edge feature transformation).
   - Total edges computation: $O(n^2 \cdot L \cdot d^2)$. There are $n^2$ edges in an MLP layer, and $L$ total layers.

2. **Aggregation (Equation 4):**
   - Per node aggregation: $O(n \cdot d)$. Each node aggregates messages from its $n$ neighbors in the graph from the previous time step.
   - Total nodes: $O(n^2 \cdot L \cdot d)$.

3. **Recurrent Memory Updates (Equation 6):** $O(n \cdot L \cdot d^2)$.

**Total Computational Complexity:** $O(n^2 \cdot L \cdot d^2 + n^2 \cdot L \cdot d + n \cdot L \cdot d^2)$

**Analysis:** For large input networks, the term $O(n^2 \cdot L \cdot d^2)$ dominates the overall computational cost.

## E.2 Space Complexity of Our Method

1. **Node and Edge Features:** $O(n^2 \cdot d + n \cdot d)$. We store the node and edge features of the graph from the previous time step, corresponding to the previous MLP layer.

2. **Memory (Equation 6):** $O(n \cdot d)$.

**Total Space Complexity:** $O(n^2 \cdot d + n \cdot d)$

# F Neural Networks as Dynamic Neural Graphs

## F.1 Additional Components in CNNs as Dynamic Neural Graphs

In Section 4.2, we outlined how to convert the fundamental modules of CNNs - the convolutional layers and the linear layers - into modules within the dynamic neural graph. In modern CNNs, besides convolutional and linear layers, there are often additional components like the flattening layer and residual connections He et al. (2016). To ensure our dynamic neural graph framework is applicable to a wide range of CNN architectures, we define how to convert flattening layers and residual connections to the modules in dynamic neural graphs.

**Flattening layer.** The flattening layer in CNNs is used to convert multi-dimensional feature maps into a single feature vector that can be accepted by fully connected layers for subsequent operations. The dimensions of the feature map output by a convolutional layer at the $l$-th layer of a CNN are $h_{ft}^l \times w_{ft}^l \times c^l$, where $h_{ft}^l$ and $w_{ft}^l$ represent the spatial dimensions of the feature map, $c^l$ denotes the number of channels of the feature map. Assuming we flatten the feature map to a single feature vector of dimension $d^l$, where $d^l = h_{ft}^l \times w_{ft}^l \times c^l$. A linear layer is then utilized at the $(l+1)$-th layer of the CNN to perform a linear transformation on this flattened feature vector, yielding a vector of dimension $d^{l+1}$.

In the dynamic neural graph, $c^l$ corresponds to the number of nodes in $\mathbf{v}^l$, and $d^{l+1}$ corresponds to the number of nodes in $\mathbf{v}^{l+1}$. We can conceptualize the function of the flattening layer as generating $h_{ft}^l \times w_{ft}^l$ virtual vertices within each node in $\mathbf{v}^l$, and they are subsequently connected to nodes $\mathbf{v}^{l+1}$. Virtual vertices do not contain any feature information, they are only used to indicate the connection relationship between their carriers $\mathbf{v}^l$ and nodes $\mathbf{v}^{l+1}$. Each connection between a virtual vertex and a node in $\mathbf{v}^{l+1}$ corresponds to a single weight scalar in the weight matrix $W^{l+1}$ of the linear layer at the $(l+1)$-th layer. In essence, this implies that each node in $\mathbf{v}^l$ is linked to a node in $\mathbf{v}^{l+1}$ via $h_{ft}^l \times w_{ft}^l$ edges by utilizing virtual vertices embedded within itself. In addition, to maintain consistency in the number of edges between each pair of nodes throughout the entire CNN, we employ the same method as proposed in Section 4.2 to pad the edges between $\mathbf{v}^l$ and $\mathbf{v}^{l+1}$. To provide a more intuitive understanding of the process of converting the flattening layer to the dynamic neural graph, we present an example in Figure 7.

**Residual Connections.** Residual connections are used in neural networks to address the gradient vanishing problem. Specifically, a residual connection in a neural network allows the input to bypass one or more layers and be added directly to the output. If a residual connection is established between the output of the $l$-th layer and the output of the $(l+r)$-th layer in a CNN, then within the corresponding dynamic neural graph, we define events occurring at timestamp $t^{l+r}$ as the addition of nodes $\mathbf{v}^l$ at the $l$-th layer, in addition to the addition and deletion of nodes and edges as defined in Section 4.2. Additionally, we add new edges to ensure that each node $\mathbf{v}_i^l$ at the $l$-th layer is connected to node $\mathbf{v}_i^{l+r}$ at the $(l+r)$-th layer via a single edge. Considering the potential differences brought by residual connections with different time spans or layer spans to the update of target nodes, we define the edge feature of each of these edges between $\mathbf{v}^l$ and $\mathbf{v}^{l+r}$ as $\mathbf{e}_{res,i}^{l+r} = \theta_{res} r$, where $\theta_{res} \in \mathbb{R}^{d_e}$ is a learnable vector.

## F.2 Transformers as Dynamic Neural Graphs

In the Transformer, the core modules are multi-head self-attention layers. Assuming there are $h$ heads in a multi-head self-attention layer. For each head $head_i$, the input $\mathbf{X}$ with a dimension of $d_{\text{model}}$ [2] is firstly

---

[2]In general, $\mathbf{X} \in \mathbb{R}^{L \times d_{\text{model}}}$. We omit the sequence length $L$ here for clarity.

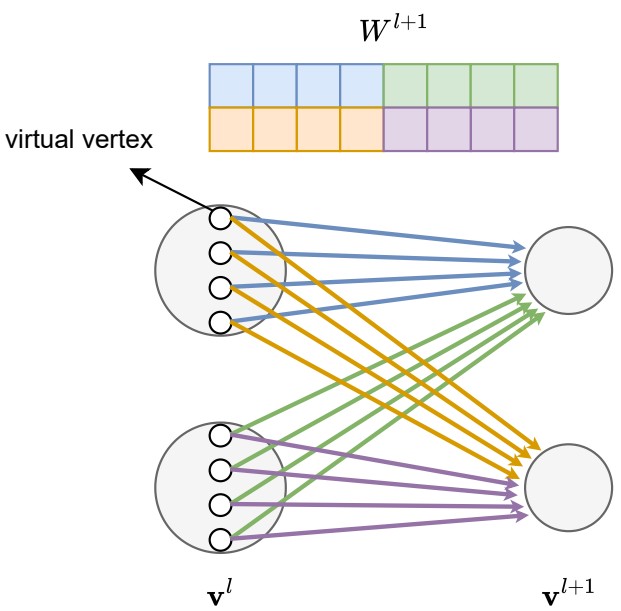

Figure 7: An example for converting the flattening layer to the dynamic neural graph, where edges correspond to the weight scalar of the same color.

transformed into $\mathbf{Q}_i \in \mathbb{R}^{d_k}$, $\mathbf{K}_i \in \mathbb{R}^{d_k}$ and $\mathbf{V}_i \in \mathbb{R}^{d_v}$ through three linear projections.

$$\mathbf{Q}_i = \mathbf{X}\mathbf{W}_i^Q, \tag{30}$$

$$\mathbf{K}_i = \mathbf{X}\mathbf{W}_i^K, \tag{31}$$

$$\mathbf{V}_i = \mathbf{X}\mathbf{W}_i^V, \tag{32}$$

where $\mathbf{W}_i^Q \in \mathbb{R}^{d_{\text{model}} \times d_k}$, $\mathbf{W}_i^K \in \mathbb{R}^{d_{\text{model}} \times d_k}$ and $\mathbf{W}_i^V \in \mathbb{R}^{d_{\text{model}} \times d_v}$. The scaled dot-product attention is then computed using $\mathbf{Q}_i$, $\mathbf{K}_i$ and $\mathbf{V}_i$, producing $\mathbf{Z}_i \in \mathbb{R}^{d_v}$.

$$\mathbf{Z}_i = \text{Attention}(\mathbf{Q}_i, \mathbf{K}_i, \mathbf{V}_i) = \text{Softmax}\left(\frac{\mathbf{Q}_i\mathbf{K}_i^\top}{\sqrt{d_k}}\right)\mathbf{V}_i \tag{33}$$

Finally, the $\mathbf{Z}_i$ from all heads are concatenated, and the output $\mathbf{Y}$ with a dimension of $d_{\text{model}}$ is produced through a linear transformation $\mathbf{W}^O \in \mathbb{R}^{hd_v \times d_{\text{model}}}$.

To convert a multi-head self-attention layer into a dynamic neural graph while still simulating its forward pass process, we divide the multi-head self-attention layer into three timestamps within the dynamic neural graph.

In the first timestamp, we simulate the linear transformation from $\mathbf{X}$ to $\mathbf{Q}_i$, $\mathbf{K}_i$ and $\mathbf{V}_i$. We begin by adding $d_{\text{model}}$ nodes to the graph, with each node representing a dimension of the input $\mathbf{X}$. Next, for each head, we add $d_k + d_k + d_v$ nodes initialized as zero vectors, corresponding to the dimensions of $\mathbf{Q}_i$, $\mathbf{K}_i$ and $\mathbf{V}_i$. Additionally, we add each element of the weight matrices $\mathbf{W}_i^Q$, $\mathbf{W}_i^K$ and $\mathbf{W}_i^V$ as a single edge to the graph, connecting the corresponding nodes. Thus, for all heads, we add $h \times (d_k + d_k + d_v)$ nodes and $h \times (d_{\text{model}} \times d_k + d_{\text{model}} \times d_k + d_{\text{model}} \times d_v)$ edges.

In the second timestamp, we simulate the computation process of scaled dot-product attention. First, we delete the nodes corresponding to the input and the edges corresponding to $\mathbf{W}^Q$, $\mathbf{W}^K$ and $\mathbf{W}^V$, keeping only the nodes representing $\mathbf{Q}$, $\mathbf{K}$ and $\mathbf{V}$. It can be observed from Equation 33 that the dot product of $\mathbf{Q}_i$ and $\mathbf{K}_i^\top$, the division by $\sqrt{d_k}$ and the element-wise multiplication with $\mathbf{V}_i$ are parameter-free operations. Inspired by the method proposed by Kofinas et al. (2024), we design a simple graph structure to fit these

operations. Specifically, for each head, we augment the graph by adding $d_v$ nodes initialized with zero vectors, corresponding to the dimensions of $\mathbf{Z}_i$. Additionally, we add edges connecting each newly added node to the nodes corresponding to $\mathbf{Q}_i$, $\mathbf{K}_i$ and $\mathbf{V}_i$. These edges are defined as learnable vectors, allowing them to fit the parameter-free operations mentioned above during training. For all heads, we add a total of $h \times d_v$ nodes and $h \times (d_k \times d_v + d_k \times d_v + d_v \times d_v)$ edges in this timestamp.

In the last timestamp, we simulate the process of mapping the concatenation of $\{\mathbf{Z}_1, \mathbf{Z}_2, ..., \mathbf{Z}_h\}$ to the output $\mathbf{Y}$. The process of performing the linear transformation using $\mathbf{W}^O$ is the same as the linear transformation using a linear layer in an MLP. Therefore, we can use the same method employed to convert linear layers into dynamic neural graphs in MLPs to transform this linear transformation process.

**Experiment.** We conduct an experiment to predict the generalization of transformers by processing their parameters, thereby validating the effectiveness of our proposed DNG framework in handling transformer architectures. To construct the dataset, we follow the approach of Small CNN Zoo Unterthiner et al. (2020) and create a transformer-specific dataset. Specifically, we train 1,000 Vision Transformer (ViT) models, each initialized differently, using SimpleViT from the ViT_pytorch library to classify the CIFAR-10 dataset. Each model is trained up to a certain epoch before being stopped, after which its parameters and test accuracy are recorded. We split the trained models into 80% for training, 10% for validation, and 10% for testing.

Similar to the settings used for CNN generalization prediction in Section 8.2, we employ our proposed DNG-Encoder followed by an MLP to predict the test accuracy of a ViT model given its parameters as input. Using the approach proposed above, we construct the DNG for transformer architectures in our experiments. The ViT model consists of two transformer blocks (10 timestamps), an embedding layer (1 timestamp), and an output layer (1 timestamp), resulting in a dynamic graph with a total of 12 timestamps. For comparison, we benchmark our DNG-Encoder against NG-GNN and NG-T (Kofinas et al., 2024). Following the official guidelines in their work, we reproduced these baselines for Transformer neural graphs. To ensure a fair comparison, we aligned the model capacities so that all methods have approximately 0.25M parameters. The computational efficiency (Peak Memory and Latency) is profiled based on the inference cost of a single sample, averaged over 10 independent runs. For predictive performance, we report the Kendall rank correlation coefficient ($\tau$) calculated as the mean and standard deviation over 3 independent runs. Table 8 shows that our method achieves the highest Kendall rank correlation coefficient ($\tau$) while demonstrating massive efficiency gains—completely avoiding the $O(N^2)$ global attention explosion seen in NG-T and achieving the fastest inference latency. These results validate the effectiveness and scalability of the DNG framework in handling complex Transformer architectures.

Table 8: Performance and efficiency comparison for the Transformer generalization prediction task ($\sim$0.25M Params).

| Method | Kendall's $\tau$ | GFLOPs | Peak Mem (MB) | Latency (ms) |
|---|---|---|---|---|
| NG-GNN | 0.8844±0.006 | 3.22 | 54.49 | 5.94 |
| NG-T | 0.8917±0.002 | 44.69 | 446.95 | 11.77 |
| **DNG-Encoder (Ours)** | **0.9028**±0.005 | **0.12** | **25.36** | **4.38** |

# G   Details of Experimental Setup

Below, we provide additional detailed explanations of the experiments outlined in Section 8.

## G.1   Classify INRs with INR2JLS

### G.1.1   Datasets

We applied the INR2JLS framework to classify images from the open-source MNIST, Fashion MNIST, and CIFAR-10 datasets as proposed by Zhou et al. (2024a). The INRs in these datasets are structured as three-layer MLPs with a hidden dimension of 32, utilizing the sine function as the activation function. These MLPs, employing the sine activation function, are commonly known as SIRENs Sitzmann et al. (2020). Following the strategy of splitting the datasets proposed by Zhou et al. (2024a), the datasets were split into

45,000 (MNIST, CIFAR) or 55,000 (FashionMNIST) training images, 5,000 validation images, and 10,000 test images. The training set is augmented by training 10 additional copies of SIRENs with different initializations for each training image, and each validation and test image has a single SIREN. Furthermore, we generate the CIFAR-100 INRs dataset following the methodology used by Zhou et al. (2024a) for generating the CIFAR-10 INRs dataset. Specifically, we train three-layer SIRENs with a hidden dimension of 32 for 5000 steps, employing the Adam optimizer with a learning rate of 5e-5. Additionally, we also train 10 additional copies of SIRENs with different initializations for each training image, while each image in the validation set and test set retains a single SIREN.

### G.1.2 Models

In the dynamic neural graph, the Fourier size of each node feature and edge feature is set to 128, and the Fourier scale is 3. In the DNG-Encoder of the INR2JLS framework, both the Message Function and the GRU have hidden dimensions of 512. In the Latent Generator, each spatial vector has a dimension of 512, and each output latent vector has a dimension of 128.

When no image augmentation is applied, we use 49 spatial vectors to generate 49 latent vectors for the MNIST and Fashion MNIST datasets. These latent vectors are then reshaped into a feature map with dimensions $7 \times 7 \times 128$. For the CIFAR-10 and CIFAR-100 datasets, 64 spatial vectors are used to generate 64 latent vectors, which are reshaped into a feature map with dimensions $8 \times 8 \times 128$. With image augmentation, which includes rotation and flipping, we employ $49 \times 6 = 294$ spatial vectors to generate 294 latent vectors for the MNIST and Fashion MNIST datasets. These vectors are reshaped into a feature map with dimensions $7 \times 7 \times (128 \times 6) = 7 \times 7 \times 768$. For the CIFAR-10 and CIFAR-100 datasets, we use $64 \times 6 = 384$ spatial vectors to generate 384 latent vectors, which are reshaped into a feature map with dimensions $8 \times 8 \times (128 \times 6) = 8 \times 8 \times 768$.

For the reconstruction task, we employ two transposed convolutional layers as a decoder, both layers with the kernel size of 4, the stride of 2, and the padding of 1 for all four datasets. The out channels of the first transposed convolutional layer are 256, the out channels of the second transposed convolutional layer are 1 (MNIST, Fashion MNIST) or 3 (CIFAR-10, CIFAR-100). The total number of parameters of the model is $5M$ for the MNIST and Fashion MNIST datasets and $6.1M$ for the CIFAR-10 and CIFAR-100 datasets.

For the classification task, we fix the trained DNG-Encoder and Latent Generator from the reconstruction task and use them for generating feature maps. For the MNIST and Fashion MNIST datasets, we utilize a classifier comprising two convolutional layers followed by a three-layer MLP. The classifier has the hidden dimension of 256, with the dropout rate of 0.5 applied between each layer. For the CIFAR-10 and CIFAR-100 dataset, the classifier structure maintains the same as above, except for adjusting the dropout rate between convolutional layers to 0.2. The total parameters of the model is $5M$ for MNIST and Fashion MNIST, $6.7M$ for CIFAR-10 and CIFAR-100.

### G.1.3 Training

For the reconstruction task, we set the training batch size to 64 and use the Adam optimizer with a learning rate of 1e-4. The model is trained for 400,000 steps, and we apply an early stopping strategy that choosing the model showing the best performance on the validation set, as described by Zhou et al. (2024a).

For the classification task, the training batch size is set to 128, and we use the AdamW optimizer with a learning rate of 1e-4. The model is trained for 200,000 steps, and the early stopping strategy is also employed.

### G.2 Predicting CNN Classifier

We use DNG-Encoder to generate the memory of the nodes in the last layer at the last timestamp of the input CNNs, and then we map the memory of these nodes to a scalar value through an MLP head, representing the test accuracy of our prediction. For DNG-Encoder, we use a multi-head Message Function introduced in Section 5 to process the information that generates the dynamic neural graph converted from CNNs. In the model, the hidden dimensions of the Message Function and GRU are set to 128. The dimension of each head of the Message Function is 32. The MLP head is a three-layer MLP, and its hidden dimension is 1024.

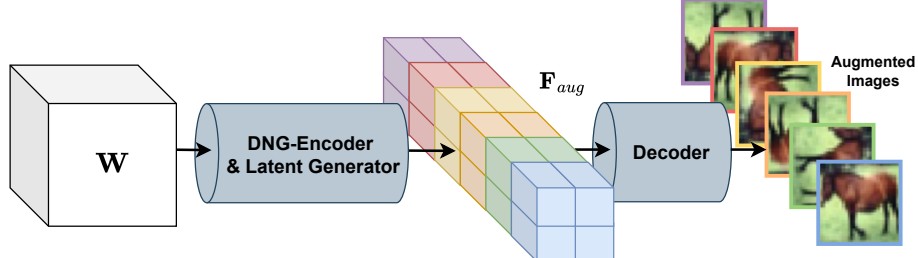

Figure 8: Data augmentation (rotation and flipping) for the INR2JLS framework.

We set the training batch size to 128, use Adam as the optimizer with learning rate of 1e-4, train for 200 epochs, and use early stopping.

### G.3 Data augmentation for INR2JLS

Data augmentation for images often introduces various image transformations during training. This allows the model to learn features and patterns under diverse conditions, enhancing its adaptability to input images and improving overall model generalization. In our proposed data augmentation method for the INR2JLS framework, we aim to enrich the feature map outputted by the DNG-encoder with a broader range of features and patterns present in the original image. This enhancement subsequently boosts the generalization ability of the classification model based on the feature map. Specifically, we rotate and flip the image corresponding to each INR, generating multiple images for each INR. Our model then learns a feature map $\mathbf{F}_{aug}$ with increased channels from the given INR to reconstruct multiple images. For a given input INR, we generate more latent vectors by defining more distinct spatial vectors to fuse with the same node memory $\mathbf{s}(t^L)$, and then permute these new latent vectors to generate a feature map $\mathbf{F}_{aug}$ with more channels. The rotation and flipping of each image in the dataset essentially alter the spatial arrangement of the pixels of the image, reflecting a same spatial arrangement of the input coordinates of each INR. After obtaining $\mathbf{s}(t^L)$ containing semantic information of the INR, we use different spatial vectors to simulate various permutations of input coordinates, and fuse them with $\mathbf{s}(t^L)$ in the latent space to decode the image. This process simulates the INR use different permutations of input coordinates to output images with different pixel arrangements. While the feature map $\mathbf{F}_{aug}$ can reconstruct distinct images to some extent, they inherently encapsulate a variety of features and patterns from these images. Consequently, the classifier can achieve better generalization when performing classification tasks based on $\mathbf{F}_{aug}$. Figure 8 shows how to employ data augmentation in the INR2JLS framework.

We also propose a data augmentation method of adding noise to the image to compare with the above proposed method of rotating and flipping the image. For the method of rotating and flipping the image, we apply five transformations on the image, including clockwise rotations of 90, 180 and 270 degrees, as well as horizontal and vertical flips, which consistent with the settings in Section 8.1. For the method of adding noise, we add Gaussian noise with a mean of 0 and a standard deviation of 0.06 the original image.

