# OpenReview forum: "Dynamic Neural Graph Encoding of Inference Processes in Deep Weight Space"
_TMLR — Accepted by TMLR_

### Review · Reviewer_cPrT · 2026-01-22

**Summary Of Contributions:**

By conceptualizing neural parameters as dynamic graphs, the authors are able to capture the temporal evolution of data during a forward pass. They process these graphs using a specialized RNN-GNN architecture that mimics sequential inference. This approach leads to INR2JLS, a method for mapping INR weights into a unified latent space to enhance downstream application performance.

**Additional Comments:**

No addional comments

**Audience:**

No

**Audience Explanation:**

Yes, I think the paper's findings would be beneficials for those individuals:

People whom interested in Parameter-Space Learning: There is a strong interest in understanding the "geometry" of weight spaces and how to treat neural networks themselves as data points.
and Implicit Neural Representations (INRs): As INRs become a standard for signals like 3D shapes and images, the TMLR community is actively looking for efficient ways to perform downstream tasks (classification, retrieval) directly on these weights via joint latent spaces.

**Broader Impact Concerns:**

Despite the improved performance metrics on CIFAR-100-INR, the core methodology—utilizing dynamic graphs to represent neural weights—presents an incremental step rather than a fundamental shift in the field. I am concerned that the novelty is limited, as the approach largely adapts established sequential modeling techniques to the parameter-space domain.

**Claims And Evidence:**

No

**Claims Explanation:**

\section*{Summary}
The paper proposes a \textbf{Dynamic Graph Neural Network} framework that treats neural network weights as nodes and edges. While existing research in this domain typically models parameters as static networks where forward propagation is limited to immediate neighbors, the authors argue that this formulation results in an \textbf{ill-posed inverse problem}. To mitigate this, they employ dynamic graphs to model propagation as time-dependent networks, utilizing an \textbf{RNN-based architecture} to embed the weights. Finally, the authors introduce \textbf{INR2JLS}, which incorporates input images during the decoding process.

\section*{Critique}
Despite the extensive mathematical derivations, I find the proposed method to be largely \textbf{incremental}. The transition from static to dynamic graphs, while technically sound, does not appear to provide a sufficient leap in conceptual \textbf{novelty} over existing parameter-space GNNs.

\section*{Questions}
\begin{enumerate}
    \item \textbf{On the Inverse Problem:} Could you please elaborate on why isolating $b_i^2$ from $(b_i^2 + W_i^2 b_i^1)$ specifically creates an ill-posed inverse problem in this context?
    \item \textbf{On Dynamic vs. Static Graphs:} Why is a dynamic graph necessary for this task? Could the same temporal dependencies not be captured using more efficient variants of Graph Neural Networks, such as \textbf{APPNP} (Approximate Personalized PageRank), which are designed to handle long-range propagation?
\end{enumerate}

**Requested Changes:**

I recommend that the authors investigate the potential for simplifying the architecture by explicitly addressing the coupling of $b_i^2$ and $W_i^2 b_i^1$. It is possible that this relationship could be modeled more efficiently through polynomial approximations, potentially removing the need for a complex dynamic formulation. Furthermore, I would like to see a comparative analysis between the proposed RNN-based GNN and a standard multi-hop propagation scheme (e.g., higher-order GNNs). Demonstrating whether multi-hop spatial message passing can capture the same dependencies would clarify if the sequential overhead of an RNN is truly necessary.

---

> ### Author Response · Authors · 2026-04-12
>
> We appreciate the reviewer’s comprehensive feedback. We respectfully argue that the transition from static to dynamic graphs represents a fundamental paradigm shift rather than an incremental improvement, as it resolves a previously unrecognized but fatal theoretical bottleneck in weight space learning: the Inverse Problem. Our response clarifies why this is essential for effective modeling and addresses concerns regarding spatial propagation, approximations, and complexity.
>
> ### 1. Addressing Concerns Regarding Conceptual Novelty
>
> Our primary innovation lies in the structural simulation of the forward pass by resolving the ill-posed inverse problem. This transition is a necessary paradigm shift and structural redesign to address the following three aspects:
>
> ### 1.1 The Fundamental Goal: Zero Information Loss
>
> In neural networks, parameters define a function $f(x)$ not merely as a collection of static values, but through their collective execution during the forward pass. The essence of weight-space learning lies in capturing this functional identity. Although a meta-model receives static weights as input, the true signal it must internalize is the computational process defined by these weights. Therefore, the model should possess the architectural capacity to approximate this sequential inference to avoid information loss. While prior work focuses on static spatial topology, our key innovation is introducing dynamic modeling, which enables the model to more faithfully simulate the forward pass—an essential prerequisite for capturing the network’s functional identity.
>
> ### 1.2 Resolving the Inverse Problem
>
> As detailed in Sec. 3.3 and Figure 1, our transition to dynamic graphs is what resolves the inverse problem that limits static methods. In a static framework, synchronous updates prematurely contaminate a layer-2 node's state with a mixed signal ($\mathbf{b}_i^2 + \mathbf{W}_i^2 \mathbf{b}^1$) at the first step. When the true activation arrives at step 2, the model must disentangle the original bias from this sum—an ill-posed inverse problem, since the individual components are no longer separately accessible. This bottleneck is highly analogous to the cross-term interference in additive positional encoding in early Transformers. By identifying this limitation, DNG provides a structural solution that avoids state contamination by enforcing causal forward computation.
>
> ### 1.3 Non-trivial Structural Innovations for Complex Architectures
>
> Our innovation extends to complex architectures like CNNs and Transformers through tailored structural optimizations. For instance, our multi-head aggregation mechanism for CNNs ensures a precise simulation of the forward pass. These architecture-specific designs yield significant gains in performance and efficiency, demonstrating that DNG is a rigorous structural redesign rather than a generic graph expansion.
>
> Collectively, these theoretical insights are validated in Sec. 7 and App. A, where DNG maintains near-zero fitting error and superior efficiency (Tables 5 & 8) by eliminating redundant computations. By resolving a fundamental paradigm flaw with a scalable structural innovation, we provide a significant, non-incremental contribution to the field.
>
> ### 2. Limitations of Static Multi-hop and Polynomial Approximations
>
> The reviewer suggested that multi-hop propagation (e.g., higher-order GNNs) or polynomial approximations (e.g., APPNP) could capture similar dependencies while simplifying the architecture. We clarify that these methods are insufficient to replace our structural modeling for the following reasons:
>
> - Causal Order vs. Spatial Distance (Higher-order GNNs): Weight space learning requires modeling causal execution, not just propagation distance. Static multi-hop schemes rely on synchronous updates that aggregate initial, untransformed states of predecessors. This fails to capture sequential non-linearities and only spreads polluted mixed-states further across the graph without resolving the inverse problem.
>
> - Linear Diffusion vs. Non-linear Execution (APPNP/ChebNet): Linear polynomial filters represent diffusive processes that cannot simulate the step-by-step non-linear transformations of a ReLU network. Furthermore, global coefficients fail to disentangle instance-specific interference terms ($W_i^2 b^1$) that vary drastically across different input networks, whereas DNG adaptively tracks these dynamic couplings.
>
> - Efficiency vs. Redundancy (RNN Overhead vs. NG-GNN): While the reviewer expressed concern over RNN overhead, static models like NG-GNN perform redundant global updates across all nodes at every step to reach deep layers. DNG's layer-wise execution updates only relevant nodes per timestamp, structurally eliminating this waste and achieving superior accuracy and efficiency by avoiding the heavy redundancy inherent in static multi-layer propagation.

---

### Review · Reviewer_kmCB · 2026-03-12

**Summary Of Contributions:**

The paper presents a dynamic neural graph encoder, which facilitates the study of neural network weight spaces.
The paper argues that dynamic graphs, which evolve over time, are superior to static graphs for network weight modelling tasks.

**Additional Comments:**

The source code comes with an environment.yml file, which documents software dependencies. While this facilitates reproduction. This projects impact would probably increase if measures were taken to improve replication in other settings.
Doing so would entail packaging the code ( https://packaging.python.org/en/latest/tutorials/packaging-projects/ ) and adding systematic tests to ensure correctness i.e. via pytest and nox ( https://docs.pytest.org/en/stable/, https://nox.thea.codes/en/stable/ ).

**Audience:**

Yes

**Audience Explanation:**

To the best of my knowledge this is a bit of a nice topic, however some individuals in TMLR's audience may be interested.

**Claims And Evidence:**

Yes

**Claims Explanation:**

The theoretical summarizes how static and dynamic neural graphs can be used to approximate neural networks. An RNN based approach is used to fit the graph encoder to the network.

The experimental section compares the static to the dynamic approach, it reports, that the dynamic approach produces a better fit to MLPs with up to three layers.
Furthermore the proposed INR2JLS method is compared to four competing methods, the authors observed improved or competitive performance.
In addition to an ablation study the paper presents results on the accuracy prediction problem on the CNN-Zoo dataset.
The experimental section concludes with an efficiency analysis and results on the effects of positional encodings and non-linearities, which complement the overall picture.

**Requested Changes:**

The acronyms INR2JLS and INR are not explained in the abstract.
Page 7 mentiones downstram tasks. What kind of downstream tasks are we talking about?
Page 9 explains this, it might be worthwhile to tell reades already on page seven.
Page 9 also refers to ` the four aforementioned datasets`, is it possible to repeat them there for extra clarity?

---

> ### Author Response · Authors · 2026-04-12
>
> We sincerely thank you for the review. Your constructive feedback has helped us significantly improve the clarity and replicability of our work. Below, we detail how we have incorporated your suggestions into the revised manuscript and supplementary materials.
>
> ### 1. Defining Acronyms in the Abstract
>
> We sincerely thank you for pointing this out. We have carefully revised the abstract to explicitly define both acronyms upon their first appearance. Specifically, we have defined INR as **Implicit Neural Representations (INRs)** and our proposed framework INR2JLS as **Implicit Neural Representation to Joint Latent Space (INR2JLS)**, ensuring complete clarity for all readers right from the beginning.
>
> ### 2. Specifying Downstream Tasks Early
>
> We appreciate this helpful suggestion, as specifying the exact task earlier greatly improves the paper's readability. We have updated the text on page 7 to explicitly replace the general term "downstream tasks" with the specific task. The revised text now reads: "...extract robust features for **INR classification**." Furthermore, we updated the subsequent sentence to state that existing methods have made great progress "in extracting representations from INRs for **classification tasks**." This ensures readers have a concrete understanding of the specific application early in the section.
>
> ### 3. Listing Dataset Names for Clarity
>
> Thank you for this excellent suggestion. To improve clarity and save readers from having to flip back through the pages to find the context, we have explicitly listed the names of the datasets in the revised manuscript. Specifically, the sentence on page 9 has been updated to: "...on the test sets of the four aforementioned datasets (MNIST, FashionMNIST, CIFAR-10, and CIFAR-100)."
>
> ### 4. Code Packaging and Automated Testing
>
> We sincerely thank you for this highly constructive suggestion regarding the engineering and replicability of our project. Following your advice, we have updated the supplementary repository by formally packaging the code using a `pyproject.toml` file according to PyPA guidelines.
>
> Additionally, we have introduced a comprehensive suite of `pytest`-based unit tests that systematically verify the correctness of our core components (e.g., graph construction, forward-pass shapes, and gradient flows) without requiring heavy computational resources. We believe these engineering improvements significantly enhance the robustness, standard compliance, and usability of our framework for the broader community.

---

> > ### Comment · Reviewer_kmCB · 2026-04-15
> > **Thank you for answering my questions**
> >
> > Thank you for answering my questions an for cleaning up the code.

---

### Review · Reviewer_AzD8 · 2026-03-30

**Summary Of Contributions:**

This paper studies learning in deep weight space by arguing that existing graph-based weight-space methods are limited by their static treatment of neural network parameters. The core idea is to represent a neural network as a dynamic neural graph, where graph snapshots evolve along the layer-by-layer forward process, and then to process this graph with a recurrent Dynamic Neural Graph Encoder (DNG-Encoder). On top of this encoder, the paper proposes INR2JLS, which maps INR weights into a joint latent space with the original image, avoiding direct weight reconstruction and instead reconstructing image content. The paper further provides experiments on (i) fitting layer-wise activations, (ii) INR classification, and (iii) CNN / transformer generalization prediction. Empirically, the strongest results are on INR classification, where the method substantially outperforms prior baselines, especially on CIFAR-10 and CIFAR-100.

Strengths

1. The paper is well motivated. The discussion of why static neural graphs may fail to faithfully simulate multi-layer forward propagation, especially through the “inverse problem” argument, gives a concrete reason for introducing dynamic graphs rather than presenting the design as a purely heuristic modification. Whether or not one fully buys the formalism, the intuition is clear and technically nontrivial.

2. The paper has a strong empirical showing on the INR classification benchmark. Table 1 reports large gains over NFN, INR2ARRAY, NG-GNN, and NG-T, with especially notable improvements on CIFAR-10 and CIFAR-100; the ablations in Table 2 also suggest that the latent-generator / image-reconstruction design matters materially for performance.

3. The method appears architecturally broader than some prior weight-space models. Beyond MLP-style INRs, the paper discusses dynamic graph construction for CNNs and transformers, and provides experiments on CNN zoo generalization prediction as well as a transformer generalization setting. Even though these experiments are smaller in scale than the INR section, they help support the claim that the framework is not confined to a single architecture family.

4. I appreciated the paper’s connection to the broader weight-space-learning perspective. In the terminology of the recent survey A Survey of Weight Space Learning: Understanding, Representation, and Generation, this submission sits primarily at the intersection of weight space representation and weight space understanding, since it both learns embeddings over weights and explicitly reasons about symmetry/equivariance in weight space. That framing helps clarify why this paper matters beyond the INR benchmark alone.

**Audience:**

Yes

**Audience Explanation:**

I expect this paper to interest readers working on neural functionals, implicit neural representations, graph representations of neural weights, and more broadly weight-space learning.

**Broader Impact Concerns:**

I do not see major immediate ethical concerns beyond the standard considerations for model analysis and weight-space inference. That said, methods that infer model properties directly from weights may eventually have dual-use implications in settings involving model attribution, capability inference, checkpoint triage, or reverse-engineering of proprietary models. A brief broader-impact discussion would strengthen the paper.

**Claims And Evidence:**

Yes

**Claims Explanation:**

The paper does provide meaningful evidence for the central claim that dynamic neural graphs are useful representations of neural network weights. The INR results are convincing, the activation-fitting motivation is coherent, and the appendices strengthen the symmetry/equivariance discussion.

However, I do not think the current evidence fully supports the strongest version of the paper’s broader claims. In particular, the empirical story is most convincing for INR classification, but more mixed elsewhere. On CNN generalization prediction, the method is slightly better than NG-T on CIFAR-10-GS but underperforms NFN(HNP) on SVHN-GS. The transformer experiment is promising, but the comparison is only against a flattened-parameter MLP baseline, which is too weak to fully establish state of the art or even strong competitiveness in that setting.

**Requested Changes:**

1. The paper should better calibrate its claims outside the INR setting. The current wording sometimes suggests broadly superior performance, but the CNN results are mixed and the transformer experiment uses only a relatively weak baseline. I would like to see either softened claims or stronger comparisons.
2. The paper should more clearly disentangle the contribution of dynamic graph encoding from the contribution of INR2JLS as a training objective / representation-learning framework. The ablations are useful, but I still found it difficult to isolate how much of the gain comes from the dynamic graph itself versus the joint latent-space supervision and augmentations.
3. The paper would benefit from an explicit efficiency/scalability discussion. Since one motivation is handling deep weight spaces, I expected runtime, memory, or scaling comparisons against static-graph baselines and sequence/token-based alternatives.
4. The related-work discussion should be broadened to better situate the paper in the larger weight-space-learning literature. I especially recommend citing and discussing:
    - W2T: LoRA Weights Already Know What They Can Do, which highlights the importance of structured/canonical weight representations for behavior prediction from weights alone.
    - A Survey of Weight Space Learning: Understanding, Representation, and Generation, which provides a useful taxonomy for placing the present work in the WSL landscape.


5. The paper would benefit from copy-editing. There are many grammatical issues and awkward phrasings that make some otherwise interesting arguments harder to follow.
6. The “inverse problem” discussion is the conceptual centerpiece of the paper and should be sharpened further, perhaps with a cleaner toy example or a more formal proposition in the main text.
7. It would be helpful to discuss more explicitly when a dynamic representation is expected to help, and when simpler static or tokenized weight representations may suffice.

---

> ### Author Response · Authors · 2026-04-12
>
> We sincerely thank the reviewer for the constructive feedback. Our responses are provided below. The following experiments have been incorporated into the revised manuscript, and the corresponding source code has also been provided.
>
> ### Q1: Stronger comparisons
> To further demonstrate the generalizability of the DNG framework, we conducted several new experiments on predicting CNN and Transformer generalization, benchmarking against static graph methods (NG-GNN, NG-T [Kofinas et al., 2024]).
>
> 1. Heterogeneous CNN Generalization: To provide a more rigorous CNN evaluation, we tested our dynamic graph method on the challenging CNN Wild Park dataset, which contains highly heterogeneous CNN architectures (varying layers, channels, skip connections, etc.). We aligned our model's parameter count ($\sim$ 0.4M) with the NG baselines. As shown below, our method significantly outperforms the static graph baselines:
>
> | Method      | Kendall's $\tau$ |
> | ----------- | ---------------- |
> | NG-GNN      | 0.8040           |
> | NG-T        | 0.8170           |
> | DNG-Encoder | 0.8743           |
>
> Note that although NFN(HNP) performs well on SVHN-GS, NFN is inapplicable in this setting, due to the heterogeneous architectures present in the dataset. We have updated the Section 8.2 by including these new results.
>
> 2. Weak Transformer experiments baseline & Efficiency: To address this concern, we carefully reproduced the NG-GNN and NG-T baselines for Transformers based on the official guidelines in their paper (Appendix C.4: "Transformers As Graphs"). To ensure a strictly fair comparison, we aligned the model capacities so that all three methods have exactly $\sim$ 0.25M parameters (we slightly reduced our original parameter count, but our performance remained robust).
> Furthermore, we report the computational efficiency (GFLOPs, Peak Memory, and Latency averaged over 10 runs) for processing these Transformer neural graphs:
>
> | Method      | Kendall's $\tau$ | GFLOPs | Peak Mem (MB) | Latency (ms) |
> | ----------- | ---------------- | ------ | ------------- | ------------ |
> | NG-GNN      | 0.8844           | 3.22   | 54.49         | 5.94         |
> | NG-T        | 0.8917           | 44.69  | 446.95        | 11.77        |
> | DNG-Encoder | 0.9028           | 0.12   | 25.36         | 4.38         |
>
> It can be observed that, under this comparison, our method not only outperforms the stronger baseline in predictive accuracy but also demonstrates clear efficiency advantages for complex Transformer graphs. We have updated the Appendix F.2 by including this new results.
>
> ### Q2:  Disentangling DNG and INR2JLS contributions
> We sincerely thank the reviewer for this suggestion. Disentangling the individual contributions is indeed important for understanding our method. The contribution of the INR2JLS framework (joint latent-space supervision and augmentations) has already been isolated in the Ablation Study (Table 2, bottom) of the original manuscript, where we compare the pure DNG-Encoder with the full INR2JLS pipeline.
>
> To specifically quantify the gain from the dynamic graph, we conducted an additional INR classification experiment comparing the pure DNG-Encoder with static graph-based classifiers (NG-GNN and NG-T [Kofinas et al., 2024]). For a fair comparison, we used only the graph encoders with a standard MLP classification head and removed the INR2JLS objective. We also excluded probe features from NG-GNN and NG-T, as these dynamic activations inject inference-time information and partially bypass the limitations of static graph structures. This setup isolates the intrinsic capability of dynamic versus static graph modeling. The table below shows the comparision results.
>
> | Method      | MNIST | Fashion | CIFAR-10 |
> | ----------- | ----- | ------- | -------- |
> | NG-GNN      | 79.60 | 71.10   | 43.94    |
> | NG-T        | 83.43 | 72.13   | 44.69    |
> | DNG-Encoder | 96.60 | 78.40   | 54.00    |
>
> The strong performance in this task futher reflects the inherent architectural advantages of DNG.
>
> ### Q3:  Efficiency Discussion
> We thank the reviewer for this suggestion. We have already provided an efficiency analysis in Sec. 9.4 (Efficiency Analysis) and Table 5. Specifically, Table 5 compares running time, memory usage, and computational complexity for processing a single INR. The results show that INR2JLS is significantly faster than prior methods, with slightly higher memory usage than NG-GNN but substantially lower than NFN and NFT, while also achieving the lowest computational complexity overall.

---

> > ### Author Response · Authors · 2026-04-12
> >
> > ### Q4:  Inverse Problem discussion
> >
> > Thanks for the suggestion. We added a scalar toy example in Sec. 3.3 to clarify the inverse-problem discussion. The example shows that the node state becomes a compound value $v^2(1)=b^2+w^2 b^1$, and recovering the correct forward-pass output requires isolating $b^2$ from this mixed term. Since $w^2 b^1$ is not separately available and varies across networks, this recovery is ill-posed. This clarifies why static representations lead to an inverse problem.
> >
> >
> >
> > ### Q5:  Dynamic vs. Static/Tokenized Applicability
> > DNG is essential for discriminative tasks on deep networks (e.g., classification, generalization prediction) because it captures deep semantics and resolves the inverse problem. As demonstrated in Section 7, the ability of static networks to simulate the forward pass progressively degrades as the input network's depth increases, which severely limits their expressiveness. While static graphs or tokenized representations perform adequately on shallow networks where information mixing is minimal, as networks get deeper, the inverse problem accumulates in static structures, and their structures incur increasingly massive computational overhead. We have incorporated this analysis into the "Discussion and Limitations" section of the revised manuscript.
> >
> > ## Other Concerns
> > - Related Work: We have expanded the Related Work to incorporate the highly relevant W2T and WSL surveys.
> > - Copy-editing: We have double-checked our manuscript and fixed several grammatical errors. Please let us know if you have any specific concerns regarding the text.
> > - Broader Impact: We have updated the Conclusion section to discuss ethical concerns and potential dual-use risks associated with our work.

---

### Decision · Action_Editor_AwqD · 2026-05-25

**Recommendation:** Accept as is

**Audience:**

Yes

**Audience Explanation:**

The paper will be of interest to at least some TMLR readers, especially those working on weight-space learning, neural functionals, implicit neural representations, and graph-based representations of neural network parameters. The topic is specialized but relevant to a growing line of work in this area.

**Claims And Evidence:**

Yes

**Claims Explanation:**

All reviewers recommend acceptance. They generally found the paper’s claims to be sound and well supported by the experimental results. In particular, the rebuttal addressed the main concerns, around the motivation for dynamic graph modeling and comparisons to static graph baselines. The added experiments improved the clarity and empirical support for the proposed method. Overall, I recommend acceptance, while noting that the reviewers expressed mixed views on whether the paper has sufficient novelty and significance for the J2C track.